# A GRID WORLD AGENT WITH FAVORABLE INDUCTIVE BIASES

## ABSTRACT

We present a novel experiential learning agent with causally-informed intrinsic reward that is capable of learning sequential and causal dependencies in a robust and data-efficient way within grid world environments. After reflecting on state-of-the-art Deep Reinforcement Learning algorithms, we provide a relevant discussion of common techniques as well as our own systematic comparison within multiple grid world environments. Additionally, we investigate the conditions and mechanisms leading to data-efficient learning and analyze relevant inductive biases that our agent utilizes to effectively learn causal knowledge and to plan for rewarding future states of greatest expected return.

## 1 INTRODUCTION

Grid world environments come in many forms and have been studied extensively in the history of Artificial Intelligence with some notable examples such as Wumpus World (Bryce, 2011), Minigrid (Chevalier-Boisvert et al., 2024), and Tileworld (Pollack & Ringuette, 1990). However, the creation of intelligent grid world agents capable of learning effectively and in a data-efficient way has posed significant challenges. Current Reinforcement Learning (RL) agents struggle in some instances due to sequential dependencies, partial observability (Wang et al., 2023a), continual learning (primacy bias Kim et al. (2024), stability-plasticity dilemma Anand & Precup (2024)), relatively high-dimensional state spaces, compared to more traditional RL tasks – even when a single value per cell is provided rather than per pixel – and sometimes non-deterministic effects of actions.

Sequential dependencies usually raise the training data demand exponentially depending on the combinatorics, and, ultimately, the number of the arising options, unless the agent is capable of learning causal representations that transfer well because chaining them is favorable for reaching an intended outcome. Partial observability, on the other hand, may require the model to have access to information from prior states, which typically corresponds to the previously observed values in a grid world outside the agent's current field of view. In terms of input dimensionality, there is a trade-off between the observation window being too small for learning an effective policy and the agent's observation window being more high-dimensional, thereby demanding more training data, even though this might be the least problematic. Additionally, if the agent is able to represent values outside of its observation window, a learned policy needs to consider not only the observation window itself but also how it spatially relates to the remembered information beyond it.

With these considerations in mind, we introduce Non-Axiomatic Causal Explorer (NACE), a novel experiential learning agent, which leverages causal reasoning and intrinsic reward signals to enable more efficient learning as well as possesses learning mechanisms with the involved inductive biases. NACE is designed to induce causal rules from temporal and spatial local changes in the grid, which are often (but not always) caused by the agent. It utilizes these rules to plan for and reach future states of maximum uncertainty in order to effectively learn more (causal rules) about the environment, thereby improving predictability-based intrinsic reward formulations.

To illustrate the effectiveness of our approach, we provide a comprehensive discussion of state-of-the-art Deep Reinforcement Learning (DRL) techniques as well as our own systematic comparison within multiple grid world environments demonstrating our agent's remarkable improvement in data efficiency, achieving similar performance with about 1000 samples where DRL algorithms typically require 1 million samples, representing a 1000-fold reduction in data requirements. We also thoroughly investigate the conditions and mechanisms under which learning in grid world environments

can become more data-efficient and attempt to answer the question about which inductive biases can lead to close-to-optimal learning speeds in the case where the agent is not pre-equipped with particular interaction rules between grid cell types but has the inductive biases to build and use them effectively. Lastly, we analyze useful inductive biases applicable among a wide range of grid worlds and their generality to raise the data efficiency of learning in such domains.

## 2 RELATED WORK

Current RL techniques such as value-based (e.g., Deep Q-Learning (Mnih et al., 2013)) and policy gradient-based (e.g., Proximal Policy Optimization (Schulman et al., 2017)) typically require millions of training iterations to solve grid-world environments (Zhang et al., 2020b), struggling to capture causal dependencies necessary for efficient planning and transferability.

Exploration improvements, such as intrinsic rewards based on information gain (Zhao et al., 2023), prediction errors (Burda et al., 2018), or visitation counts (Zheng et al., 2021; Wang et al., 2023b), enhance sample efficiency but often lack structured reasoning for generalization. In contrast, Tsividis et al. (2021) propose Theory-Based RL (TBRL), exemplified by EMPA (Exploring, Modeling, and Planning Agent), which integrates Bayesian causal modeling, structured exploration, and heuristic planning to generalize efficiently across tasks with minimal training. Similarly, GALOIS (Cao et al., 2022) addresses generalization by synthesizing interpretable, hierarchical programs with strict cause-effect logic, though its reliance on predefined program sketches limits flexibility in loosely structured environments. As a model-based RL alternative, DreamerV3 (Hafner et al., 2023) learns latent state dynamics and improves behavior through imagination, enabling generalization across diverse tasks with minimal domain-specific adjustments. However, its reliance on learning both latent representations and their dynamics reduces sample efficiency compared to methods that assume predefined representations.

Symbolic approaches like STRIPS and Behavior Trees (BTs) (Guo et al., 2023; Colledanchise & Ögren, 2018) handle human-defined causal knowledge but lack adaptive learning capabilities. Similarly, POMDPs (Spaan, 2012) focus on probability updates rather than causal discovery, while causal networks (Pearl, 1995) and structure-learning methods (Zheng et al., 2018) face challenges with ambiguity and scalability.

While TBRL, GALOIS, and DreamerV3 provide solutions for structured or model-based learning, their reliance on predefined logic, priors, or latent space learning introduces additional complexity. Our approach proposes lightweight "empirical" causal relations learned from recurring cause-effect patterns, supporting efficient real-time learning in grid worlds without predefined program sketches, Bayesian modeling, or latent space learning. This ensures adaptability while retaining interpretability and transferability.

### 2.1 SELECTED RL TECHNIQUES IN GRID WORLDS

DRL often struggles with sample efficiency, requiring substantial interactions with environments. This paper examines foundational algorithms, scalable architectures, and exploration-focused methods that address these challenges.

Foundational methods include Deep Q-Network (DQN) Mnih et al. (2015), which combines deep neural networks with Q-learning to handle large state spaces but struggles with sparse rewards; Advantage Actor-Critic (A2C) Mnih et al. (2016), which reduces variance in updates through synchronized parallel actors but is limited by its on-policy nature; Trust Region Policy Optimization (TRPO) Schulman et al. (2015), which ensures stable policy updates with trust region constraints but is computationally intensive; and Proximal Policy Optimization (PPO) Schulman et al. (2017), which refines TRPO with clipped objectives for improved data utilization and computational efficiency.

Scalable architectures such as Importance Weighted Actor-Learner Architectures (IMPALA) Espeholt et al. (2018) address multi-task learning by leveraging distributed architectures with off-policy corrections, offering scalability but facing synchronization challenges, whereby exploration-focused methods aim to address sparse rewards and complex state spaces. Count-Based Exploration (COUNT) Bellemare et al. (2016) uses pseudo-counts for better exploration but is computationally

demanding in large state spaces. Random Network Distillation (RND) Burda et al. (2018) incentivizes novelty through prediction errors, however, it depends on high-quality state representations. Curiosity-Driven Exploration (CURIOSITY) Pathak et al. (2017) rewards prediction errors of action outcomes, fostering intrinsic motivation, while Rewarding Impact-Driven Exploration (RIDE) Raileanu & Rocktäschel (2020) focuses on impactful actions but may struggle with ambiguous state changes. Adversarially Motivated Intrinsic Goals (AMIGO) Campero et al. (2021) generates adversarial goals to guide exploration, requiring robust goal-generation mechanisms for effectiveness.

Furthermore, model-based RL techniques can learn a model of the environment and improve their behavior through imagined future scenarios Sekar et al. (2020). This framework is particularly advantageous in high-dimensional and variable state spaces. DreamerV3 Hafner et al. (2023) builds on this principle by introducing a general algorithm designed to address a diverse range of tasks with minimal domain-specific adjustments. By learning latent state dynamics and planning through imagination, DreamerV3 offers broad applicability while reducing the need for extensive tuning or specialized configurations. However, the additional complexity of learning latent space representations together with their dynamics leads to less sample efficiency compared to when representations are already present and only the dynamics need to be learned.

## 3  NON-AXIOMATIC CAUSAL EXPLORER

**NACE** is our proposed experiential learning technique with causality-informed intrinsic reward and strong inductive biases for grid world environments to boost sample efficiency. Here, we provide formal descriptions of NACE. For a comprehensive list of symbols used, refer to Appendix C.

### 3.1  STATES AND RULE REPRESENTATION

State in NACE is a tuple $s = (s_{\text{spatial}}, s_{\text{internal}})$ consisting of a two-dimensional value array $s_{\text{spatial}} \in \mathbb{N}^{m \times n}$ and a one-dimensional value array $s_{\text{internal}} \in \mathbb{N}^k$, as shown in Figure 1. The two-dimensional array reflects the spatial structure in the grid world, including remembered cells beyond the current view, while the one-dimensional array is used for internal values, such as inventory items (e.g., keys).

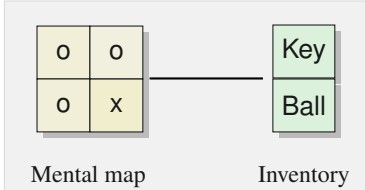

Figure 1: State components

Each rule is of the form $(preconditions, action) \Rightarrow consequence$ where the precondition can hold a conjunction of cell value constraints spatially relative to the cell value of the consequence, and the consequence predicts one particular cell's value as well as the values of the one-dimensional array at the next timestep as depicted in Figure 2. Examples of created rules are provided in Appendix G.

| $\overline{c}^1_{t-1}$ | (e.g. consequence cell = empty) | | |
|---|---|---|---|
| $\overline{c}^2_{t-1}$ | (e.g. right of consequence cell = agent ) | $\overline{c}_t$ | (e.g. consequence cell = agent) |
| $\overline{c}^k_{t-1}$ | (cell k value constraint, e.g. omitted) | $\overline{v}_t$ | (e.g. still holding key) |
| $\overline{v}_{t-1}$ | (value array constraint, e.g. holding key) | $R(r)$ | (reward predicted by rule, e.g. 0) |
| $\overline{a}_{t-1}$ | (taken action constraint, e.g. move left) | | |

Figure 2: Rule schema

Each rule tracks evidence using counters for $w_+$ and $w_-$ similar to (Wang, 2013), which measure the accuracy of the rule's predictions. Positive evidence ($w_+$) is accumulated whenever a perfectly matching rule predicts correctly, while negative evidence ($w_-$) increases with incorrect predictions. Tracking of evidence helps the agent refine its causal knowledge by prioritizing more reliable rules.

## 3.2 INDUCTIVE BIASES

It is well-known that favorable inductive biases can enhance sample efficiency. Below are inductive biases that are incorporated in NACE and relevant for many grid world environments:

1. **Temporal Locality**: NACE constructs rules based solely on the current and previous state, modeling relevant dependencies locally in time.

2. **Causal Representation**: NACE's knowledge representation is centered around the afore-mentioned causal rules which can be chained and are independent of the objective.

3. **Spatial Equivariance**: Ability to model causal dependencies between grid cells independently of the specific location of the cells considered in the dependency. This means learned rules in NACE can be applied at any location.

4. **State Tracking**: Ability to effectively track state outside of the field of view of the agent based on the recorded or estimated locations. NACE explicitly keeps track of a bird's-eye view map by recording observations into it, updating the values that are within its observability window.

5. **Attentional Bias**: Relevant dependencies tend to involve values that have either observably changed or a different value than predicted. Only rules that show a change from the previous to the current timestep, or differ from the predicted value, are considered for rule formation, evidence updating, and prediction.

Additional discussions on inductive biases as well as ablation studies can be found in Appendix B.

## 3.3 CURIOSITY MODEL

This section outlines the mechanism which helps NACE systematically acquire missing causal knowledge about the environment. The key principle is realized by making the agent plan to reach a state which it is most unfamiliar with. The familiarity is judged by whether existing rules match well to the situation, whereby matching is a matter of degree dependent on how many rule conditions match the cells in the known state. This motivates the following formalism:

- *Match* value of a rule $r$ is evaluated relative to consequence cell $c$:

$$M(r, c) = \frac{\text{Number of matched preconditions}}{\text{Total number of preconditions}}$$

- *Cell match* value of a cell $c$ dependent on all $m$ existing rule match values:

$$C(c) = \max(0, M(r_1, c), \ldots, M(r_m, c))$$

- *State match* value $S(s)$ of a state $s$ is the average $C(c)$ of its cells with $C(c) > 0$. This value, as we will see, is the secondary "explorative" objective in the planning process that guides the agent's decisions:

$$S(s) = \sum_{c \in X(s)} \frac{C(c)}{|X(s)|} \text{ where } X(s) = \{c \in s | C(c) > 0\},$$

## 3.4 NACE ARCHITECTURE

Figure 3 illustrates the high-level architecture of NACE, which consists of several interconnected components that work together to enable learning and decision-making. The related pseudocode is provided in Appendix D.

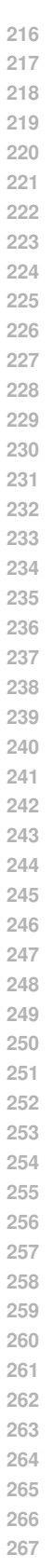

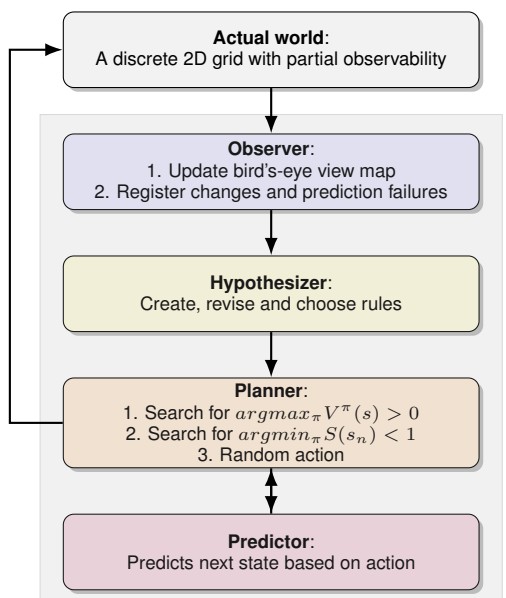

**Actual world** represents the real simulated 2D grid environment (Minigrid) with a cell-granular partial observability model. In each frame, the field-of-view local to the agent is passed on to the observer.

**Observer** takes the field-of-view 2D array as input and detects changes in values as well as identifies prediction failures from rules that predict incorrectly.

**Hypothesizer** creates and updates rules based on whether their predictions align with observations, whereby only changed-cells and prediction-mismatch cells as reported by Observer are considered.

**Planner** searches for optimal actions that lead to greater-than-zero expected return, and if none such is found, searches for actions that lead to a state of lowest state match value greater than zero. Finally, in case such also does not exist, a random action is chosen

**Predictor** forecasts the next state from the current state and the taken action, utilizing individual rules to predict a state transition of the entire state, whereby for each cell its predicted value comes from the rule with the highest $M(r, c)$.

Figure 3: Flow diagram of the system

1. **Observer:** Its role is to update a bird's-eye view map via values from the partial observation 2D array, then to find changes in input, as well as prediction-observation mismatches (prediction failures). Formally this corresponds to determining the sets:

   - **Set of changes in observations**: $M_t^{\text{change}} = \{c_{t,x,y}^{\text{observation}} \mid c_{t,x,y}^{\text{observation}} \neq c_{t-1,x,y}^{\text{observation}}\}$
     This set captures all grid cells $c_{x,y}$ where the observed value has changed between timesteps $t-1$ and $t$, highlighting areas that have been updated or modified.
   - **Set of observation mismatches**: $M_{\text{mismatched},t}^{\text{observation}} = \{c_{t,x,y}^{\text{observation}} \mid c_{t,x,y}^{\text{prediction}} \neq c_{t,x,y}^{\text{observation}}\}$
     This set includes all grid cells where the observed value differs from the predicted value at time $t$, indicating potential prediction failures.
   - **Set of prediction mismatches**: $M_{\text{mismatched},t}^{\text{prediction}} = \{c_{t,x,y}^{\text{prediction}} \mid c_{t,x,y}^{\text{prediction}} \neq c_{t,x,y}^{\text{observation}}\}$
     This set identifies all grid cells where the predicted value does not match the observed value at time $t$, from the perspective of predictions.

   These sets enable the Observer to track state changes and prediction failures, ensuring an accurate understanding of the environment and supporting the system's adaptive and predictive capabilities.

2. **Hypothesizer:** Associating positive and negative evidence based on prediction success, as well as creating new rules when positive evidence is found for the first time.
   Formally, for each rule $r = ((\bar{c}_{t-1}^1 \wedge ... \wedge \bar{c}_{t-1}^k \wedge \bar{v}_{t-1} \wedge \bar{a}_{t-1}) \Rightarrow (\bar{c}_t \wedge \bar{v}_t \wedge R(r)))$, $\bar{c} := (c_r = c)$ indicates that the value in the rule precondition aligns with the actual cell value, value array in case of $\bar{v}$, and taken action in case of $\bar{a}$.
   The rule preconditions are met when all *equality constraints* $\bar{c}_{t-1}^1, ..., \bar{c}_{t-1}^k, \bar{v}_{t-1}, \bar{a}_{t-1}$ hold. Positive evidence is attributed when the equality constraints of the postcondition $\bar{c}_t, \bar{v}_t$ are met as well and the predicted reward aligns with the observed reward ($R_t = R(r)$), where only cells which changed value or have a different value than predicted are considered to increase computational efficiency:

$$w_+(r) = \begin{cases} w_+(r) + 1 & \text{if } \{c_{t-1}^1, ..., c_{t-1}^k, c_t\} \subseteq (M_t^{change} \cup M_{\text{mismatched}_t}^{\text{observation}}) \\ w_+(r) & \text{otherwise} \end{cases}$$

   Negative evidence is assigned when any of the postcondition equality constraints are not met:

$$w_-(r) = \begin{cases} w_-(r) + 1 & \text{if } c_t \in M_{\text{mismatched}_t}^{\text{prediction}} \\ w_-(r) & \text{otherwise} \end{cases}$$

Finally, rules $r$ for which $w_-(r) > w_+(r)$ become inactive, and for two rules $r_1, r_2$ if their preconditions match (including the action) but the postconditions are different, only the rule with the higher truth expectation is selected, which is calculated according to:

$$w(r) = w_-(r) + w_+(r), \quad frequency(r) = \frac{w_+(r)}{w(r)}, \quad confidence(r) = \frac{w(r)}{w(r)+1}$$

$$f_{exp}(r) = (frequency(r) - \tfrac{1}{2}) * confidence(r) + \tfrac{1}{2}$$

This not only allows the system to find the relevant preconditions under which a consequence happens when the action is utilized but also gives the system tolerance to non-deterministic effects and enables accounting for uncertainty. A brief analysis of this can be found in Appendix A.

3. **Planner**: NACE makes use of depth- and width-bounded Breadth-First-Search algorithm with a combined search objective consisting of two components: it searches for states resulting from the different action sequences for futures that lead to the max. expected return or, if not existing, the lowest state match value. Hence, it applies a key RL principle to maximize the expected long-term return (Sutton et al., 1999), with the policy determined by the considered action sequence: $\pi(t) = a_t$ for $t = 1, 2, \ldots, n$ whereby $n$ is smaller-or-equal (dependent on where the optimum is found) to the maximum planning horizon:

$$\pi(t) = \begin{cases} \arg\max_\pi V^\pi(s_0) & \text{if } V^\pi(s_0) = \mathbb{E}\left[\sum_{t=0}^n \gamma^t R(s_t) \mid s_0 = s, \pi\right] > 0 \\ \arg\min_\pi S(s_n) < 1 & \text{otherwise} \end{cases}$$

According to this definition, if no return greater than zero can be obtained for any considered action sequence, the system instead plans for a future state of lowest state match value, whereby $(\forall t : (0 \leq t < n) \rightarrow S(s_t) = 1) \wedge S(s_n) < 1$, meaning the action sequence is constrained to be planned in such a way that state match value is 1 except for the last action where it is minimized for the resulting state.

Such constraint maximizes the agent's chance to reach the state of minimum state match while ensuring the low match value is not a consequence of predicting further from states where the knowledge was already not fully applicable.

Due to the amount of possible options, the planning algorithm dominates the asymptotics of NACE. It has the computational complexity of $O(|V| + |E|)$ where $V$ is the set of nodes, and $E$ is the set of edges of the search graph. Constant-bounded search depth and width can be achieved by pruning of branches by expected return and state match value, however, bounded search depth can negatively affect performance, as analyzed in Appendix A.

4. **Predictor**: When the planner queries for the predicted state from a given state and an action, the role of the predictor is to construct the predicted state by applying all knowledge to the given state in the following way: initializing with the cell values from the given state, where for each cell we utilize only the rule $r$ with $M(r, c) = 1$ and maximum $f_{exp}(r)$, meaning the rule preconditions match perfectly to the given state, the action that has been considered, and $r$ has the highest truth expectation among the rule candidates.

In this case the postcondition cell value of the rule is applied to the corresponding cell at position $(x, y)$ in the predicted state, while else the cell keeps the value from the previous state. Hence, for utilized rules $r^* = ((\overline{c}_t^1 \wedge \ldots \wedge \overline{c}_t^k \wedge \overline{v}_t \wedge \overline{a}_t) \Rightarrow (\overline{c}_{t+1} \wedge \overline{v}_{t+1} \wedge R(r^*)))$, where $\overline{c}_{t+1}$ and $\overline{v}_{t+1}$ constrains the cell value and value array of the consequence:

$$c_{t+1,x,y} = \begin{cases} c_{t+1} & \text{if } r^* = \underset{r \mid M(r, c_{t,x,y})=1}{\arg\max} f_{\exp}(r) \\ c_{t,x,y} & \text{otherwise} \end{cases}$$

Now, while $s_{t+1}$ is a composition of the cells at all locations at time $t + 1$, the reward associated with $s_{t+1}$ is the average of the reward of each of the $N$ utilized rules:

$$R(s_{t+1}) = \tfrac{1}{N} \sum_{i=1}^N R(r_i^*)$$

Hereby, the average was chosen since if the reward prediction of all the used rules aligns with the observed reward, their average will also align, while the sum would overestimate the outcome.

## 4 Experiments in Minigrid

To evaluate the effectiveness of NACE compared to other DRL techniques, we conducted a series of experiments in Minigrid Chevalier-Boisvert et al. (2024), a 2D grid world environment featuring diverse and procedurally generated scenarios (Hardware setup and further test environments in Appendices F and H). We focus on Minigrid levels that feature partial observability (using the default observation format, which provides values per grid cell rather than per pixel), challenging the agent to operate with limited information about its surroundings. The selected environments (see Table 1) are categorized based on the specific challenges they present:

1. **Static:** fixed start & goal locations.

2. **Dynamic:** randomized start, goal, and obstacle positions.

3. **Dynamic with sequential dependencies:** tasks requiring specific action sequences (e.g., a door that needs a key or switch to be opened).

| Environment | Type |
|---|---|
| MiniGrid-Empty-16x16-v0 | 1 |
| MiniGrid-DistShift2-v0 | 1 |
| MiniGrid-LavaGapS7-v0 | 2 |
| MiniGrid-SimpleCrossingS11N5-v0 | 2 |
| MiniGrid-Unlock-v0 | 3 |
| MiniGrid-DoorKey-8x8-v0 | 3 |

Table 1: Environments with corresponding types

In each environment, we recorded the average reward, episode length, and standard deviation every 100 timesteps, whereby each timestep incorporates the observed state, action taken, and obtained reward. The following sections present and discuss some representative results for each category, using the selected RL techniques mentioned in Section 2.1. (Configuration and hyperparameter details are in Appendix E). Additionally, Behavior Trees (BTs) and hard-coded policies are employed as performance upper bounds in non-stationary and static environments.

### 4.1 Stationary Environments

In this category, because the start and goal locations are fixed, the primary challenge for the agent is to consistently learn and optimize navigation strategies over repeated episodes.

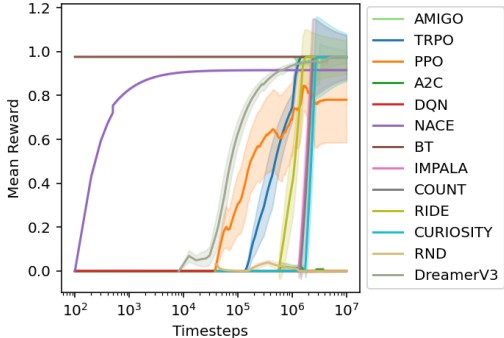

| Techn. | Avg. reward | S. dev. |
|---|---|---|
| **TRPO** | **0.976** | **0.001** |
| **PPO** | 0.781 | 0.390 |
| **A2C, DQN** | 0.000 | 0.000 |
| **IMPALA** | **0.976** | **0.197** |
| **COUNT** | 0.974 | 0.203 |
| **RIDE** | 0.975 | 0.151 |
| **CURIOSITY** | 0.974 | 0.217 |
| **RND** | 0.000 | 0.008 |
| **AMIGO** | **0.976** | **0.059** |
| **DreamerV3** | **0.976** | **0.000** |
| **NACE** | 0.916 | 0.004 |

Figure 4: Learning curves in *MiniGrid-Empty-16x16-v0*

Table 2: Final values for *MiniGrid-Empty-16x16-v0*

**MiniGrid-Empty-16x16-v0:** This environment features a large, static grid where the agent must navigate from a fixed start location to a fixed goal. Due to the limited observation window and the sparse reward—only granted upon reaching the goal—the task can present some difficulties. In this case, NACE, following its innate strategy, first learns to move effectively within the grid by exploring its immediate surroundings. Then it systematically expands its exploration, hitting observed walls out of curiosity, and finally exploring initially invisible parts of the map until the goal object is found and moved into. Due to its intrinsic inductive bias, it explores the area systematically and associates reward with the goal location within about $10^3$ timesteps. In contrast, other techniques

like DreamerV3, TRPO, IMPALA, RIDE, and AMIGO, although capable of eventually learning the task, require over $10^5$ timesteps to perform comparatively well (as seen in Figure 4, and Table 2).

**MiniGrid-DistShift2-v0:** In this case, the fixed start and goal locations are accompanied by stationary lava obstacles, which the agent must navigate around to reach the goal. DQN and DreamerV3 perform quite well, achieving a near-optimal policy with an average reward of 0.96, closely mirroring the performance of the BT. Although reaching a slightly lower average reward of 0.87, NACE was three orders of magnitudes more sample-efficient. The next-best policies were found by AMIGO and PPO with an average reward of 0.78 and 0.76, while the other techniques were below 0.5, all of them being much less sample-efficient than NACE (as in Figure 5 and Table 3).

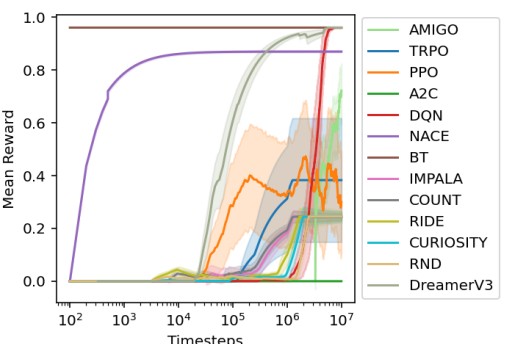

| Techn. | Avg. reward | S. dev. |
|---|---|---|
| **TRPO** | 0.383 | 0.469 |
| **PPO** | 0.763 | 0.381 |
| **A2C** | 0.000 | 0.000 |
| **DQN** | **0.961** | **0.000** |
| **IMPALA** | 0.245 | 0.027 |
| **COUNT** | 0.243 | 0.025 |
| **RIDE** | 0.245 | 0.036 |
| **CURIOSITY** | 0.245 | 0.041 |
| **RND** | 0.245 | 0.049 |
| **AMIGO** | 0.778 | 0.203 |
| **DreamerV3** | **0.961** | **0.000** |
| **NACE** | 0.870 | 0.006 |

Figure 5: Learning curves in *MiniGrid-DistShift2-v0*

Table 3: Final values for *MiniGrid-DistShift2-v0*

## 4.2 DYNAMIC ENVIRONMENTS

Given that the start and goal locations, along with obstacle positions, are randomized in each episode, these environments require the agent to continuously adapt to new and unpredictable conditions.

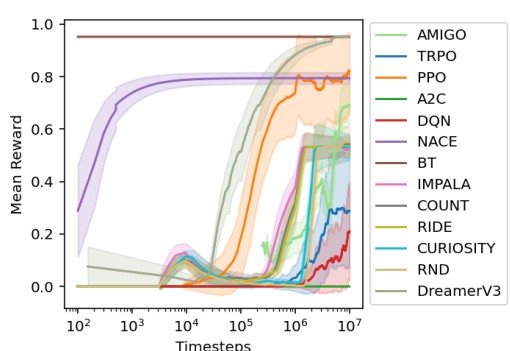

| Techn. | Avg. reward | S. dev. |
|---|---|---|
| **TRPO** | 0.187 | 0.375 |
| **PPO** | 0.838 | 0.309 |
| **A2C** | 0.000 | 0.000 |
| **DQN** | 0.114 | 0.309 |
| **IMPALA** | 0.521 | 0.064 |
| **COUNT** | 0.543 | 0.067 |
| **RIDE** | 0.535 | 0.070 |
| **CURIOSITY** | 0.531 | 0.103 |
| **RND** | 0.551 | 0.111 |
| **AMIGO** | 0.690 | 0.151 |
| **DreamerV3** | **0.952** | **0.005** |
| **NACE** | 0.794 | 0.044 |

Figure 6: Learning curves in *MiniGrid-LavaGapS7-v0*

Table 4: Final values for *MiniGrid-LavaGapS7-v0*

**MiniGrid-LavaGapS7-v0:** In this environment, the agent must navigate around randomly placed lava obstacles to reach a fixed goal, requiring adaptability due to the varying paths between episodes. The 5x5 free space, which is mostly covered by the agent's observation window, is complicated by dynamically spawning lava, unlike the stationary obstacles in *MiniGrid-DistShift2-v0*. From the mean episode rewards (as seen in Figure 6, and Table 4), it is clear that PPO and NACE find a similar effective strategy, whereby NACE takes about $10^3$ timesteps while PPO takes $3 \times 10^5$ timesteps to reach a mean reward value of around 0.8, while the optimal policies, as BT shows, are between 0.9 and 1.0, a range only DreamerV3 (0.953) managed to enter. Additionally, PPO exhibits more

instability in learning and greater sensitivity to initialization, as evidenced by a higher standard deviation. Following these, AMIGO reached an average reward of only $0.69$, while the remaining techniques performed poorly, despite the fact that this level is practically fully observable.

**MiniGrid-SimpleCrossingS11N5-v0:** Here the agent faces a large grid with multiple intersections and potential dead ends. The randomized layout in each episode also forces the agent to develop a robust exploration strategy. As Figure 7, and Table 5 show, DreamerV3, IMPALA, COUNT, RIDE, CURIOSITY, RND achieved near-optimal policies since their intrinsic reward mechanisms seem to be particularly helpful in this large environment where the observable window covers only a small part. NACE and AMIGO found reasonable policies with average rewards of $0.88$ and $0.78$ respectively, while the remaining techniques scored below $0.5$. Again, NACE's strength lies in its data efficiency, driven by its inductive biases, even though it does not converge to the optimal policy.

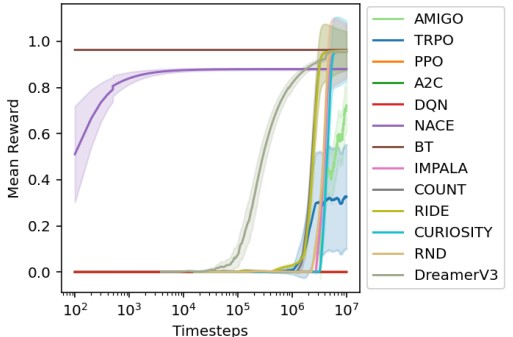

| Techn. | Avg. reward | S. dev. |
|---|---|---|
| **TRPO** | 0.381 | 0.467 |
| **PPO, A2C, DQN** | 0.000 | 0.000 |
| **IMPALA** | 0.958 | 0.238 |
| **COUNT** | **0.960** | **0.168** |
| **RIDE** | 0.959 | 0.170 |
| **CURIOSITY** | 0.958 | 0.261 |
| **RND** | 0.958 | 0.222 |
| **AMIGO** | 0.778 | 0.203 |
| **DreamerV3** | **0.960** | **0.008** |
| **NACE** | 0.880 | 0.009 |

Figure 7: Learning curves in *MiniGrid-SimpleCrossingS11N5-v0*

Table 5: Final values for *MiniGrid-SimpleCrossingS11N5-v0*

### 4.3 DYNAMIC ENVIRONMENTS WITH SEQUENTIAL DEPENDENCIES

In these environments, the need to perform actions in a specific sequence adds complexity and tests the agent's ability to plan and execute multi-step strategies.

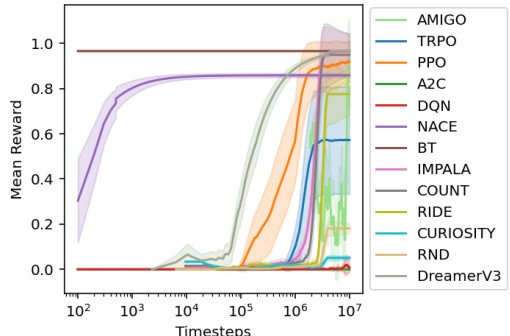

| Techn. | Avg. reward | S. dev. |
|---|---|---|
| **TRPO** | 0.577 | 0.471 |
| **PPO** | 0.890 | 0.263 |
| **A2C, DQN** | 0.000 | 0.000 |
| **IMPALA** | 0.964 | 0.162 |
| **COUNT** | 0.949 | 0.185 |
| **RIDE** | 0.775 | 0.188 |
| **CURIOSITY** | 0.051 | 0.016 |
| **RND** | 0.181 | 0.046 |
| **AMIGO** | 0.932 | 0.388 |
| **DreamerV3** | **0.967** | **0.003** |
| **NACE** | 0.858 | 0.018 |

Figure 8: Learning curves in *MiniGrid-Unlock-v0*

Table 6: Final values for *MiniGrid-Unlock-v0*

**MiniGrid-Unlock-v0:** In this scenario, the agent must first locate and pick up a key before unlocking a door to reach the goal and obtain the reward. This sequential dependency adds a layer of complexity that challenges the agent's ability to plan ahead. Even though it is a single sequential dependency, the DRL techniques that learned the fastest initially, DreamerV3 and PPO, demands almost a million timesteps to converge to a similarly effective policy as NACE, which achieves this within just $10^3$ steps again (as seen in Figure 8, and Table 6). Additionally, while PPO shows more instability in learning, it is far less chaotic than AMIGO. IMPALA reached the optimal policy, and

did so after about 2 million steps, performing similarly well as COUNT and AMIGO in the end. It is also visible, in our runs, that TRPO did not exceed a mean episode reward of $0.6$, while A2C and DQN completely failed to learn any effective policy.

**MiniGrid-DoorKey-8x8-v0:**  This environment introduces an additional layer of sequential dependency by requiring the agent to navigate through an unlocked door to reach a goal in a separate room. While passing through the door adds complexity, the primary challenge lies in the sparse reward structure, as no reward is given for merely using the door, since only reaching the final goal is rewarded. As presented in Figure 9, Table 7, DreamverV3 and COUNT nearly achieved the optimal policy with an average reward of $0.975$ and $0.96$. AMIGO reached $0.87$, but within $10^7$ timesteps, which is below the average reward which NACE reached within only $10^4$ steps. Overall the results suggest poor combinatorial scaling of the involved DRL techniques, while NACE, on average, required a similar amount of timesteps as for *MiniGrid-Unlock-v0* to learn an effective policy.

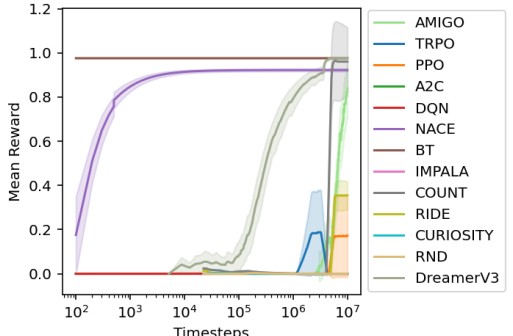

| Techn. | Avg. reward | S. dev. |
|---|---|---|
| **TRPO** | 0.000 | 0.000 |
| **PPO** | 0.156 | 0.357 |
| **A2C, DQN** | 0.000 | 0.000 |
| **IMPALA** | 0.000 | 0.000 |
| **COUNT** | 0.960 | 0.308 |
| **RIDE** | 0.354 | 0.126 |
| **CURIOSITY** | 0.000 | 0.000 |
| **RND** | 0.000 | 0.001 |
| **AMIGO** | 0.868 | 0.241 |
| **DreamerV3** | **0.977** | **0.004** |
| **NACE** | 0.922 | 0.012 |

Figure 9: Learning curves in *MiniGrid-DoorKey-8x8-v0*

Table 7: Final values for *MiniGrid-DoorKey-8x8-v0*

The observed sample efficiency of NACE originates from explicitly exploiting the cell-based grid world state observations for creating transition rules. This represents a strong inductive bias, which makes NACE less generic than DreamerV3. However it can nevertheless be valuable in broader applications where mapping high-dimensional input to a similar discrete world representation is feasible. This mapping, dependent on the problem domain, can be implemented with the appropriate choice of feature extraction techniques. Such approaches are commonly employed in robotics, where methods like Simultaneous Localization and Mapping and object detection models are integrated to construct semantic maps for operating mobile robots Zhang et al. (2020a). However we acknowledge this demands a considerable engineering effort, while DreamerV3 can update its perceptual representations dynamically via gradient-based updates. Additional discussions, such as about NACE's sub-optimality due to representational limitations, can be found in Appendix A.

## 5 CONCLUSION

We introduced NACE, an experiential learning agent designed to enhance data efficiency in grid world environments by leveraging causally-informed intrinsic rewards and inductive biases. We compared NACE with state-of-the-art DRL techniques, demonstrating that while these techniques are able to eventually achieve near-optimal policies, they often require significantly more data, especially as task complexity increases due to factors such as sequential dependencies. NACE, by contrast, extends the RL framework to empirically support causal relations, enabling effective learning and decision-making even in low data settings without relying on pre-defined causal models. Our causality-informed curiosity model, combined with the outlined inductive biases, facilitates systematic exploration and learning requiring significantly fewer timesteps. We hope that future work in the field will strike new compromises regarding the inclusion of inductive biases, leading to highly sample-efficient DRL that retains the ability to converge to optimal policies. Moving forward, we plan to generalize NACE to handle three-dimensional and continuous spaces, as well as explore neural implementations of NACE, further advancing the capabilities of learning agents.

## REPRODUCIBILITY STATEMENT

- We utilized open-source implementations of the selected DRL algorithms from public repositories (not including our technique):
    - AMIGO was from here: `https://github.com/facebookresearch/adversarially-motivated-intrinsic-goals`
    - BT is here: `https://github.com/andreneco/minigrid_bt`
    - DQN, A2C, TRPO, and PPO were established on Stable Baselines3 (SB3)'s baselines repository (Raffin et al., 2021): `https://stable-baselines3.readthedocs.io/`
    - DreamerV3 was from here: https://github.com/qxcv/dreamerv3
    - All the other were from here: `https://github.com/sparisi/cbet`
- We used the the Minigrid package for the environments in our comparison, which is available here: `https://github.com/Farama-Foundation/Minigrid`
- For NACE we provide a stand-alone zip archive for reviewers to reproduce our results, which is runnable on a regular computer with Python interpreter. It includes a README.txt in the NACE folder, as well as scripts to generate the tables and the plots present in the paper.

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

## APPENDIX A  ADDITIONAL STUDIES AND DISCUSSIONS

The performance results indicate that NACE often exhibits sub-optimal outcomes. To analyze this, we present contributing factors from a conceptual perspective, examine the impact of hyperparameter choices, and assess robustness to non-determinism arising from random action consequences.

- **Representational Limitations**: NACE's rule-based framework captures only spatially relative dependencies from one timestep to the next. It does not exploit the inherent structural statistics of environment generation, which are leveraged by various DRL techniques. While these structural dependencies are most apparent in static environments where locations remain constant, they are also present in dynamic environments. For example, the goal location consistently appears in the bottom-right corner not only in *MiniGrid-Empty* levels but also in *MiniGrid-SimpleCrossingS11N5-v0*. NACE's inability to utilize these broader environmental patterns limits its performance compared to methods that can.

  Additionally, NACE's rules are tied to an action, meaning agent-external changes that are not caused by NACE need to be learned for each action separately, considerably lowering its sample efficiency by a factor of the amount of actions. The mechanism could be extended to incorporate learning of rules without an action as a precondition, leaving it to evidence collection whether the action is considered dependent on the truth expectation of the alternative rules.

- **Study of Reduced Planning Horizon**: NACE's estimation of expected returns relies heavily on its planning horizon. Short planning horizons can significantly reduce performance, especially in tasks requiring long-term planning. To quantify this effect, we examine two cases: *MiniGrid-DoorKey-8x8-v0*, which demands longer-horizon planning, and *MiniGrid-DoorKey-6x6-v0*, which is less demanding in this regard. As shown in Table 8, running NACE with a planning horizon of only 8 steps in *MiniGrid-DoorKey-8x8-v0* results in convergence to an average return of 0.48, whereas extending the horizon to 100 steps improves the average reward to 0.92. In contrast, in *MiniGrid-DoorKey-6x6-v0*, NACE maintains an average reward of 0.93 regardless of the planning horizon. A similar pattern is observed in *MiniGrid-Empty-16x16-v0*, where the average reward drops from 0.91 to 0.41 when the planning horizon is reduced from 100 to 8 steps. These results highlight NACE's dependence on adequate planning horizons for effective rule chaining and the significant performance degradation that occurs when the planning horizon is too short.

| Environment | Planning Horizon, Average Reward |
|---|---|
| MiniGrid-DoorKey-6x6-v0 | 8 steps, 0.93 |
| MiniGrid-DoorKey-6x6-v0 | 100 steps, 0.93 |
| MiniGrid-DoorKey-8x8-v0 | 8 steps, 0.48 |
| MiniGrid-DoorKey-8x8-v0 | 100 steps, 0.92 |
| MiniGrid-Empty-16x16-v0 | 8 steps, 0.41 |
| MiniGrid-Empty-16x16-v0 | 100 steps, 0.91 |

Table 8: NACE Performance with different planning horizons

- **Robustness to Non-Determinism**: NACE's rule representation incorporates uncertainty handling through evidence counters, enabling it to cope with non-deterministic state transitions. To assess this capability, we modify the environment to invoke unintended actions with certain probabilities. In *MiniGrid-Empty-16x16-v0*, when 10% of actions result in unintended outcomes, NACE still achieves an average reward above 0.9, demonstrating basic tolerance to non-determinism. However, when the probability of unintended actions increases to 20%, NACE fails to complete the task within the maximum allowed time in all episodes. Higher tolerance to non-determinism can be achieved by increasing the default truth expectation threshold for rule usage above the default value of 0.5. However, this adjustment reduces sample efficiency, as it requires the agent to confirm each rule multiple times before utilizing it.

# APPENDIX B  ABLATION STUDY: EFFECTS OF OMITTING KEY INDUCTIVE BIASES IN NACE, AND INDUCTIVE BIASES IN DRL

## B.1  CAUSAL RULE REPRESENTATION

The causal rule representation is foundational to NACE's operation and cannot be omitted. However, we analyze the effects of reducing the planning horizon, which limits the depth of chaining, in Appendix A.

## B.2  TEMPORAL LOCALITY AND ATTENTIONAL BIAS

Omitting these biases with larger environment sizes is infeasible due to the combinatorial explosion of potential rules as we will now analyze.

- For an environment of size $w \times h$, the number of possible rule preconditions for a single timestep is $2^{w \cdot h}$, as each particular cell can either be considered or not be considered in the precondition of a rule.

- For a time window of duration $d$, this expands to $2^{w \cdot h \cdot d}$, leading to 18446744073709551616 possible rule preconditions for an $8 \times 8$ grid within a single timestep.

- NACE is tied to Markov Assumption particularly within the observational window, as all rule construction and updating considers only the previous and current state. However, its bird view map representation also contains values from observations of previous time steps which are currently out of view of the agent, which brings us to the next point, state tracking.

## B.3  STATE TRACKING

Without state tracking, NACE lacks memory of prior observations and memory of observations which are outside of its field-of-view. This results in oscillatory behavior caused by the exploration strategy of the agent, as it can only utilize information the visible information.

- In our experiments across 10 runs in MiniGrid-Empty-8x8-v0, this led the agent to turn indefinitely due to the curiosity model assigning low match values to previously visited areas (due to the lack of state tracking they are always considered to be of unknown value)

which are now outside of the field-of-view of the agent. The closest such cell is immediately behind the agent with the default partial observation model in Minigrid, which explains the behavior.

- State Tracking plays a critical role in ensuring purposeful exploration and decision-making, for the agent to know which places have been visited and what it has been observing at the particular locations, as well as which locations have yet to be observed.

- Sequential dependencies often depend on state tracking. An example of this is when a door has to be opened with a key, where the key and the door is too far apart to be observed concurrently, demanding some form of spatial memory. Another form of state tracking lies in the observable inventory array, which when absent would need the modeling of long-range temporal dependencies (e.g. did the agent already pick up the key?) which would demand a suitable model structure to be learnable by the agent.

### B.4   SPATIAL EQUIVARIANCE

The absence of spatial equivariance significantly impacts sample efficiency.

- Each rule must be learned independently for every location, meaning in an 8x8 grid, the agent has to learn 64 times the same set of rules. However, since particular arrangements of cell values will not re-appear through the environment generation, it can take significantly longer to learn the relevant knowledge without this bias.

- Hence for the general case with an environment of size $w \times h$, this increases the required sample count at least by a factor of $w \cdot h$, harming significantly the sample efficiency of the technique.

- Conceptually, we also would like to point out that the rule learning mechanisms do not allow to learn spatial equivariance retrospectively either, while some DRL techniques, dependent on the model structure, could potentially acquire it.

These results highlight the necessity of each inductive bias in ensuring the scalability, efficiency, and functionality of NACE.

### B.5   WHICH INDUCTIVE BIASES ARE PRESENT IN THE DRL TECHNIQUES

In the main paper we outlined the inductive biases of NACE. However we would like to point out that some of them are also inherent in the DRL techniques, complementing our discussion on inductive biases in DRL and NACE:

- **Temporal Locality**: The DRL methods perform best when the Markov Assumption is met, despite LSTM allowing to cope with partial observability, the need to capture long-range temporal dependencies makes sample efficient learning more difficult.

- **Causal representation**: While not explicitly stated as a set of cause-effect relations, DreamerV3's learned dynamics model can predict the consequence states of actions, which is not the case for the model-free DRL methods. Such modeling is to some extent independent from the objective (what is rewarding), and allows an agent to train itself from simulated experience by predicting novel states, and to reach novel goals.

- **Spatial Equivariance**: Clearly the DRL techniques do not have an explicit rule representation, however the Convolution layers in the DRL policies allow for learned features to be identified at different locations, improving generalization.

- **State Tracking**: Is not explicitly handled by the DRL techniques as a separate point, instead it is handled in the same way as non-local temporal dependencies in the LSTM-including policies, while NACE builds a bird view map explicitly, which can be considered to be a form of spatial memory.

- **Attentional Bias**: While NACE has a strong prior to which cells to consider based on observably changed values and prediction mismatches, the DRL policies with Convolution layers are more flexible and allow an agent to learn which values are relevant in relation to each other.

## APPENDIX C  NOTATION AND SYMBOLS

| Symbol | Description |
|--------|-------------|
| $s$ | State, represented as a combination of a 2D grid ($s_{grid}$) and a 1D array ($s_{array}$) |
| $a$ | Action taken by the agent |
| $r$ | Causal rule in the form (preconditions, action) $\Rightarrow$ consequence |
| $c_{t,x,y}$ | Cell value at position $(x, y)$ in the 2D grid at time t |
| $\overline{c}$ | Equality constraint on a cell value (e.g., $c_r = c$) |
| $v_t$ | Value array at time t |
| $\overline{v}$ | Equality constraint on value array (e.g., $v_r = v$) |
| $M(r, c)$ | Match value of a rule $r$ for cell $c$, based on the fraction of preconditions satisfied |
| $C(c)$ | Cell match value for cell $c$, derived from the maximum match value across all rules |
| $S(s)$ | State match value for state $s$, calculated as the average $C(c)$ for cells with $C(c) > 0$ |
| $w_+(r)$ | Positive evidence counter for rule $r$, incremented when predictions align with observations |
| $w_-(r)$ | Negative evidence counter for rule $r$, incremented when predictions differ from observations |
| $w(r)$ | Total evidence count for rule $r$, defined as $w(r) = w_+(r) + w_-(r)$ |
| $frequency(r)$ | Fraction of positive evidence for rule $r$, defined as $f(r) = \frac{w_+(r)}{w(r)}$ |
| $confidence(r)$ | Confidence factor for rule $r$, defined as $c(r) = \frac{w(r)}{w(r)+1}$ |
| $f_{\exp}(r)$ | Expected truth value for rule $r$, calculated as $f_{\exp}(r) = (f(r) - \frac{1}{2}) \cdot c(r) + \frac{1}{2}$ |
| $M_t^{change}$ | Set of cells with changes in observed values between timesteps $t - 1$ and $t$ |
| $M_{\text{mismatched},t}^{observation}$ | Set of cells where observed values differ from predicted values at timestep $t$ |
| $M_{\text{mismatched},t}^{prediction}$ | Set of cells where predictions differ from observations at timestep $t$ |
| $R(r)$ | Reward associated with rule $r$ |
| $R(s)$ | Reward associated with a state $s$, defined as the average reward of rules applied to generate $s$ |
| $V(s)$ | Value of state $s$, used in planning for maximizing long-term returns |
| $\pi(t)$ | Planned action sequence or policy at timestep $t$ |
| $\gamma$ | Discount factor for future rewards |

## APPENDIX D  PSEUDOCODE

The system can be described by the pseudocode:

---

Algorithm 1: Pseudocode of NACE

- **Actual World:** perceived_array = perceive_partial(world)
- **Observer:**
  $s_t$ = update_bird_view($s_{t-1}$, perceived_array)
  $calculate(M_{change}, M_{mismatched}^{observation}, M_{mismatched}^{prediction})$
- **Hypothesizer:**
  - Create new rules for which $w_+(r) = 1$.
  - Update rule evidences according to $w_+(r)$ and $w_-(r)$.
  - Choose rules $r_1$ with $w_+(r_1) > w_-(r_1)$ for which there does not exist a rule $r_2$ with same precondition and action, but different postcondition with $f_{exp}(r_2) > f_{exp}(r_1)$.
- **Planner utilizing Predictor:**
  $a_1, ..., a_n = BFS\_with\_Predictor(V(s) > 0)$
  $a_1^*, ..., a_n^* = BFS\_with\_Predictor(min(S(s)) < 1)$
  //whereby BFS_with_Predictor is bounded breadth first search with Predictor as state transition function
  If found($a_1, ..., a_n$):, return $a_1, ..., a_n$
  If found($a_1^*, ..., a_n^*$):, return $a_1^*, ..., a_n^*$
  Else Perform a random action

---

# APPENDIX E  HYPERPARAMETER DETAILS

## E.1  FOUNDATIONAL ALGORITHMS

### E.1.1  CORE MODELS AND THEIR MECHANISMS

- **Deep Q-Network (DQN):** DQN integrates deep neural networks with classical Q-learning, making it effective for handling large state spaces. To stabilize training, DQN uses experience replay and a separate target network. The Q-value update in DQN follows:

$$Q(s,a) \leftarrow Q(s,a) + \alpha \big( R(s) + \gamma \max_{a'} Q(s',a') - Q(s,a) \big)$$

  where:

  - $s, a$: current state and action,
  - $s', a'$: next state and action,
  - $R(s)$: reward received,
  - $\gamma$: discount factor for future rewards,
  - $\alpha$: learning rate.

- **Advantage Actor-Critic (A2C):** A2C builds on the actor-critic framework, synchronizing multiple parallel learners to reduce variance in policy updates. It calculates an **advantage function** to evaluate actions relative to the current policy's value estimate, stabilizing training but requiring frequent environmental interactions due to its on-policy nature.
  **Advantage Function:**

$$A(s,a) = Q(s,a) - V(s)$$

  **Policy Update:** The policy is updated using the gradient:

$$\theta \leftarrow \theta + \alpha \nabla_\theta \log \pi_\theta(a|s) A(s,a)$$

  where:

  - $Q(s,a)$: action-value function,
  - $V(s)$: state-value function,
  - $\pi_\theta(a|s)$: policy parameterized by $\theta$,
  - $\alpha$: learning rate.

- **Trust Region Policy Optimization (TRPO):** TRPO addresses stability in policy updates by enforcing a trust region constraint, ensuring small policy changes during optimization. This constraint is implemented via a KL-divergence bound, preventing drastic shifts in behavior but requiring computationally expensive second-order optimization.
  **Objective Function:**

$$\max_\theta \mathbb{E}_{s \sim \pi_{\theta_{\text{old}}}} \left[ \frac{\pi_\theta(a|s)}{\pi_{\theta_{\text{old}}}(a|s)} A(s,a) \right]$$

  **Constraint:**

$$\mathbb{E}_{s \sim \pi_{\theta_{\text{old}}}} \left[ D_{\text{KL}}(\pi_{\theta_{\text{old}}} || \pi_\theta) \right] \leq \delta$$

  where:

  - $\pi_\theta(a|s)$: new policy,
  - $\pi_{\theta_{\text{old}}}(a|s)$: previous policy,
  - $A(s,a)$: advantage function,
  - $D_{\text{KL}}$: KL-divergence,
  - $\delta$: trust region size.

- **Proximal Policy Optimization (PPO):** PPO refines TRPO by introducing a clipped surrogate objective, which simplifies computation and allows for multiple updates per batch. This approach improves data utilization while maintaining policy stability.
  **Clipped Surrogate Objective:**

$$\max_\theta \mathbb{E}_{s,a} \left[ \min \left( \frac{\pi_\theta(a|s)}{\pi_{\theta_{\text{old}}}(a|s)} A(s,a), \text{clip} \left( \frac{\pi_\theta(a|s)}{\pi_{\theta_{\text{old}}}(a|s)}, 1 - \epsilon, 1 + \epsilon \right) A(s,a) \right) \right]$$

  where:

- $\pi_\theta(a|s)$: new policy,
- $\pi_{\theta_{old}}(a|s)$: old policy,
- $A(s,a)$: advantage function,
- $\epsilon$: clipping threshold.

### E.1.2 HYPERPARAMETER CONFIGURATION FOR FOUNDATIONAL ALGORITHMS

We utilize the Stable Baselines3 framework (Raffin et al., 2021) to train and evaluate foundational algorithms, leveraging its pre-implemented models and customizable configurations. All algorithms use the same convolutional neural network architecture to process observations, ensuring consistency across experiments. The hyperparameters for each algorithm were selected based on achieving the best average performance across all tasks, rather than optimizing for a single task, to ensure generalizability. The details of the network architecture and training setup for each algorithm are outlined below.

- **Network Architecture:** Observations ($7 \times 7 \times 3$) from the Minigrid environment are processed through four convolutional layers. Each layer is configured as follows:
  - Kernel size: $2 \times 2$
  - Activation: ReLU
  - Increasing number of filters: 16, 32, 64, and 128

  The output of the final convolutional layer is flattened and passed to a fully connected layer with:
  - Output dimension: 128
  - Activation: ReLU
- **Training Setup for DQN:**
  - Learning rate: 0.0001
  - Buffer size: $1,000,000$
  - Learning starts: 100
  - Batch size: 32
  - Soft update coefficient: 1
  - Discount factor: 0.99
  - Train frequency: 4
  - Gradient steps: 1
  - Target update interval: $10,000$
  - Exploration fraction: 0.1
  - Initial exploration epsilon: 1.0
  - Final exploration epsilon: 0.05
  - Max gradient norm: 10.0
- **Training Setup for A2C:**
  - Learning rate: 0.0007
  - Number of steps: 5
  - Discount factor: 0.99
  - Entropy coefficient: 0.0
  - Value function coefficient: 0.5
  - Max gradient norm: 0.5
- **Training Setup for TRPO:**
  - Learning rate: 0.001
  - Number of steps: 2048
  - Batch size: 128
  - Discount factor: 0.99
  - Conjugate gradient max steps: 15

- Conjugate gradient damping: 0.1
- Line search shrinking factor: 0.8
- Line search max iterations: 10
- Number of critic updates: 10
- Target KL divergence: 0.01

- **Training Setup for PPO:**
  - Learning rate: 0.0003
  - Number of steps: 2048
  - Batch size: 64
  - Number of epochs: 10
  - Discount factor: 0.99
  - Clip range: 0.2
  - Entropy coefficient: 0.0
  - Value function coefficient: 0.5
  - Max gradient norm: 0.5

### E.2 MODEL-BASED ALGORITHM: DREAMERV3

DreamerV3 is a model-based RL algorithm designed to enhance sample efficiency by learning a latent world model of the environment. It optimizes both the world model and the policy within the latent space, reducing the computational demands of interacting with the environment.

**World Model:** The latent dynamics model predicts future latent states $z$ based on prior latent state $z_{t-1}$, action $a_{t-1}$, and reward $R_{t-1}$. This model facilitates long-term planning without requiring explicit rollouts in the actual environment.

**Policy Optimization:** The policy maximizes expected rewards in the learned latent space by leveraging the dynamics model to simulate trajectories. Policy updates use gradient-based methods informed by imagined rollouts.

**Loss Function:**

$$\mathcal{L}_{\text{DreamerV3}} = \mathcal{L}_{\text{Reconstruction}} + \mathcal{L}_{\text{Dynamics}} + \mathcal{L}_{\text{Policy}}$$

where:

- $\mathcal{L}_{\text{Reconstruction}}$: Measures the accuracy of reconstructing environment observations,
- $\mathcal{L}_{\text{Dynamics}}$: Captures consistency in latent state transitions,
- $\mathcal{L}_{\text{Policy}}$: Maximizes imagined rewards.

**Hyperparameter Configuration for DreamerV3:** The hyperparameter configuration has been chosen to match the settings provided in `https://github.com/qxcv/dreamerv3`. To avoid redundancy and maintain brevity, we do not include the full configuration here due to its extensive nature.

### E.3 MODEL-FREE EXPLORATION AND SCALABILITY EXTENSIONS

All experiments for the other model-free methods are based on the Torchbeast implementation of IMPALA (Espeholt et al., 2018), which has been modified to support intrinsic reward algorithms as described in Raileanu & Rocktäschel (2020) and Campero et al. (2021). The hyperparameters were selected following the configurations used in these references. For clarity, we first list the values shared across all algorithms, followed by the specific details unique to each one.

#### E.3.1 SHARED HYPERPARAMETERS

- **Network Architecture:** Observations ($7 \times 7 \times 3$ for Minigrid) are processed through three convolutional layers:
  - Number of filters: 32 per layer

- Kernel size: $3 \times 3$
- Stride: 2
- Padding: 1
- Activation: Exponential Linear Unit (ELU)

The output of the convolutional layers is passed to:

- An LSTM layer to address partial observability by maintaining temporal dependencies and encoding sequences of observations.
- A fully connected layer for computing:
  - * Policy logits: Unnormalized scores for each action, converted to probabilities using a softmax function.
  - * Value estimates: Predictions of expected future returns, used in actor-critic methods.

- **Training Setup:**

  - Number of actors: 40
  - Number of buffers: 80
  - Unroll length: 100
  - Number of learner threads: 4
  - Batch size: 32
  - Discount factor: 0.99
  - Learning rate: 0.0001
  - Policy entropy loss: 0.0005
  - Gradient clipping: Norm of 40
  - Save interval: Every 20 minutes

- **Special Parameters (Only When Applicable):**

  - Count reset probability: 0.001 (COUNT, RIDE)
  - Hash bits: 128 (COUNT)

### E.3.2 INTRINSIC REWARDS AND COEFFICIENTS

Intrinsic rewards address sparse rewards and inefficient exploration. Each algorithm applies scaling coefficients to normalize its intrinsic rewards. Additionally, all techniques incorporate policies enhanced with LSTMs to address partial observability by maintaining memory of past observations and actions.

- **IMPALA:** No intrinsic reward ($r_i = 0.0$).
- **COUNT:** $r_i = 0.005$.
- **RIDE:** $r_i = 0.1$.
- **CURIOSITY:** $r_i = 0.1$.
- **RND:** $r_i = 0.1$.
- **AMIGO:** $r_i = 0.1$ (applies to the teacher's intrinsic rewards).

The formal definitions of the intrinsic rewards are:

**COUNT:** The intrinsic reward is based on state visitation counts, encouraging exploration of less-visited states:

$$r_i = \frac{1}{N(s_0)},$$

where $N(s_0)$ is the (pseudo)count of visits to state $s_0$. Counts are never reset during training.

**RIDE (Rewarding Impact-Driven Exploration):** The intrinsic reward combines state novelty and state-change impact:

$$r_i = \|\phi(s) - \phi(s_0)\|_2 \cdot \frac{1}{N(s_0)},$$

where $\phi$ is trained to minimize both forward and inverse dynamics prediction errors. Counts $N(s_0)$ are reset at the beginning of each episode.

**CURIOSITY:** The intrinsic reward comes from the prediction error of a forward dynamics model $f$, which predicts the next state embedding $\phi(s_0)$ from the current embedding $\phi(s)$ and action $a$:

$$r_i = \|f(\phi(s), a) - \phi(s_0)\|_2.$$

**RND (Random Network Distillation):** The intrinsic reward is computed as the prediction error of a trainable network $\phi$ attempting to match the output of a fixed random network $\hat{\phi}$:

$$r_i = \|\phi(s_0) - \hat{\phi}(s_0)\|_2.$$

**AMIGO:** The teacher policy generates goals $g$ for the agent, with rewards given as:

$$r_i = v(s_t, g) = \begin{cases} +1 & \text{if } s_t \text{ satisfies } g, \\ 0 & \text{otherwise.} \end{cases}$$

The total reward is a weighted sum of intrinsic and extrinsic rewards:

$$r_t = \beta r_i + \alpha r_e, \quad \text{with } \beta = 0.3, \alpha = 0.7.$$

### E.3.3 ALGORITHM-SPECIFIC HYPERPARAMETERS AND ARCHITECTURES

- **IMPALA (Baseline):**
  - Intrinsic reward: None.
  - Loss: Policy gradient, baseline, and entropy losses.
- **COUNT:**
  - Intrinsic reward: State visitation counts.
  - Uses count reset probability: $p = 0.001$.
- **CURIOSITY:**
  - Intrinsic reward: Forward prediction error.
  - Modules:
    * State embedding model: Encodes observations into 256-dimensional embeddings.
    * Forward dynamics model: Predicts next state embedding given current embedding and action.
    * Inverse dynamics model: Predicts action given two successive state embeddings.
  - Loss weights:
    * Forward dynamics loss: 10.0.
    * Inverse dynamics loss: 0.1.
- **RIDE:**
  - Intrinsic reward: Product of state visitation counts and the norm of state embedding changes.
  - Modules: Same as CURIOSITY.
- **RND:**
  - Intrinsic reward: Prediction error between random target network and predictor network embeddings.
  - Modules:
    * Random target network: Produces fixed embeddings for observations.
    * Predictor network: Trained to predict random target embeddings.
  - Loss weight: 0.1.

- **AMIGO:**

  - Intrinsic reward: Teacher-generated rewards.
  - Teacher-specific parameters:

    * Intrinsic reward coefficient ($\beta$): 0.3.
    * Extrinsic reward coefficient ($\alpha$): 0.7.
    * Generator batch size: 150.
    * Generator entropy cost: 0.05.
    * Generator threshold: $-0.5$.

## APPENDIX F    HARDWARE AND RUNTIME

In this section, we describe the hardware setup used to run the techniques and provide runtime characteristics, including the duration of a single run for each technique. We specify the CPU and GPU types and indicate whether the GPU was utilized for the corresponding models. While we report this information for reproducibility, we emphasize that the focus of our analysis is not on computational cost, but rather on sample efficiency.

**NACE:**

- CPU: Apple M2 with 24GB RAM

- GPU: Not utilized for this technique

- Runtime: Approximately 15 minutes runtime till convergence per environment per run with respective seed.

**Intrinsic reward models (COUNT, RIDE, CURIOSITY, RND, AMIGO) and IMPALA:**

- CPU: Intel Core i7-9750H with 32GB RAM

- GPU: Geforce GTX-1660 Ti with 6GB RAM

- Runtime: Approximately 8 hours per run on average

**Baseline models (TRPO, PPO, A2C, DQN):**

- CPU: 1 compute node with 64 cores and 512GB RAM in total

- GPU: NVIDIA Tesla A100 HGX GPU with 80GB RAM

- Runtime: Approximately 1 hour per run on average

**DreamerV3:**

- CPU: 1 compute node with 64 cores and 512GB RAM in total

- GPU: NVIDIA Tesla A100 HGX GPU with 80GB RAM

- Runtime: Approximately 40 hours per run on average

## APPENDIX G   EXAMPLE ENVIRONMENT WITH LEARNED RULES

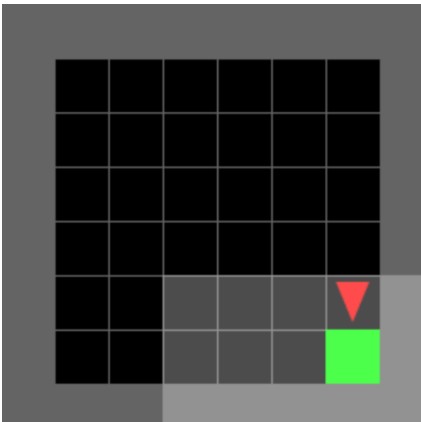

Figure 10: Illustration of Minigrid-Empty-8x8

The following are all the rules NACE learns in the Minigrid-Empty-8x8 environment as illustrated in Figure 10:

```
Agent interacting with goal location:
<(v=[1], c[ 0, 0 ]='x', c[ 0, 1 ]='H', ^down ) =/> (v=[0], c[ 0, 0 ]='.', R(r)=1)>.
<(v=[1], c[-1, 0 ]='H', c[ 0, 0 ]='x', ^left ) =/> (v=[0], c[ 0, 0 ]='.', R(r)=1)>.
<(v=[1], c[ 0,-1 ]='H', c[ 0, 0 ]='x', ^up  ) =/> (v=[0], c[ 0, 0 ]='.', R(r)=1)>.
<(v=[1], c[ 0, 0 ]='x', c[ 1, 0 ]='H', ^right) =/> (v=[0], c[ 0, 0 ]='.', R(r)=1)>.
Goal location interacting with agent:
<(v=[1], c[ 0,-1 ]='x', c[ 0, 0 ]='H', ^down ) =/> (v=[0], c[ 0, 0 ]='.', R(r)=1)>.
<(v=[1], c[ 0, 0 ]='H', c[ 1, 0 ]='x', ^left ) =/> (v=[0], c[ 0, 0 ]='.', R(r)=1)>.
<(v=[1], c[ 0, 0 ]='H', c[ 0, 1 ]='x', ^up  ) =/> (v=[0], c[ 0, 0 ]='.', R(r)=1)>.
<(v=[1], c[-1, 0 ]='x', c[ 0, 0 ]='H', ^right) =/> (v=[0], c[ 0, 0 ]='.', R(r)=1)>.
Agent interacting with empty space:
<(v=[1], c[ 0, 0 ]=' ', c[ 0, 1 ]='x', ^up  ) =/> (v=[1], c[ 0, 0 ]='x', R(r)=0)>.
<(v=[1], c[-1, 0 ]='x', c[ 0, 0 ]=' ', ^right) =/> (v=[1], c[ 0, 0 ]='x', R(r)=0)>.
<(v=[1], c[ 0,-1 ]='x', c[ 0, 0 ]=' ', ^down ) =/> (v=[1], c[ 0, 0 ]='x', R(r)=0)>.
<(v=[1], c[ 0, 0 ]=' ', c[ 1, 0 ]='x', ^left ) =/> (v=[1], c[ 0, 0 ]='x', R(r)=0)>.
Empty space interacting with agent:
<(v=[1], c[ 0,-1 ]=' ', c[ 0, 0 ]='x', ^up  ) =/> (v=[1], c[ 0, 0 ]=' ', R(r)=0)>.
<(v=[1], c[ 0, 0 ]='x', c[ 1, 0 ]=' ', ^right) =/> (v=[1], c[ 0, 0 ]=' ', R(r)=0)>.
<(v=[1], c[ 0, 0 ]='x', c[ 0, 1 ]=' ', ^down ) =/> (v=[1], c[ 0, 0 ]=' ', R(r)=0)>.
<(v=[1], c[-1, 0 ]=' ', c[ 0, 0 ]='x', ^left ) =/> (v=[1], c[ 0, 0 ]=' ', R(r)=0)>.
Agent interacting with wall:
<(v=[1], c[ 0,-1 ]='o', c[ 0, 0 ]='x', ^up  ) =/> (v=[1], c[ 0, 0 ]='x', R(r)=0)>.
<(v=[1], c[ 0, 0 ]='x', c[ 1, 0 ]='o', ^right) =/> (v=[1], c[ 0, 0 ]='x', R(r)=0)>.
<(v=[1], c[ 0, 0 ]='x', c[ 0, 1 ]='o', ^down ) =/> (v=[1], c[ 0, 0 ]='x', R(r)=0)>.
<(v=[1], c[-1, 0 ]='o', c[ 0, 0 ]='x', ^left ) =/> (v=[1], c[ 0, 0 ]='x', R(r)=0)>.
Wall interacting with agent:
<(v=[1], c[ 0, 0 ]='o', c[ 1, 0 ]='x', ^left ) =/> (v=[1], c[ 0, 0 ]='o', R(r)=0)>.
<(v=[1], c[ 0, 0 ]='o', c[ 0, 1 ]='x', ^up  ) =/> (v=[1], c[ 0, 0 ]='o', R(r)=0)>.
<(v=[1], c[-1, 0 ]='x', c[ 0, 0 ]='o', ^right) =/> (v=[1], c[ 0, 0 ]='o', R(r)=0)>.
<(v=[1], c[ 0,-1 ]='x', c[ 0, 0 ]='o', ^down ) =/> (v=[1], c[ 0, 0 ]='o', R(r)=0)>.
```

The amount of learned rules required to deal with the Minigrid environments required typically vary between 16 (minimum with walls and free space) and usually less than 100 dependent on the amount of cell types, whereby for two cell types to interact with $m$ actions, at least $2 * m$ additional rules are learned.

## APPENDIX H   TEST ENVIRONMENTS

Prior to moving to Minigrid NACE was first tested with internal levels.

- Level 1: Food collection. In this level, as depicted in Figure 11, the agent needs to collect food.

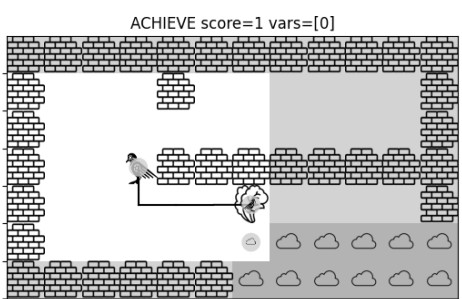

Figure 11: Food collection

- Level 2: Doors and keys. In this level, as depicted in Figure 12, the agent needs open doors with keys in order to collect batteries.

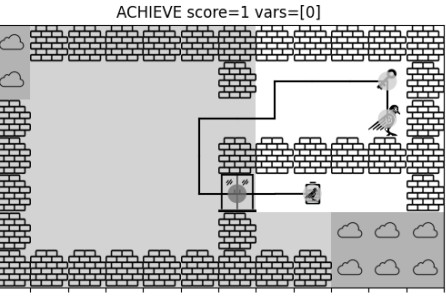

Figure 12: Food collection

- Level 3: A pong game in a grid world as illustrated in Figure 13, where the agent can only move vertically and needs to catch the ball by predicting its movement.

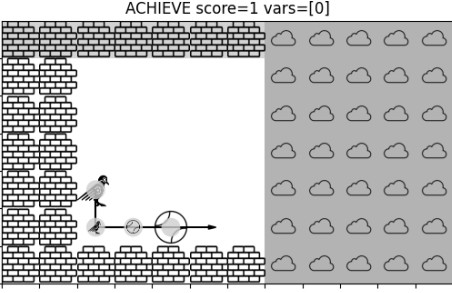

Figure 13: Pong game

- Level 4: Egg delivery. In this level, as depicted in Figure 14, the agent needs to deliver eggs to the chicken.

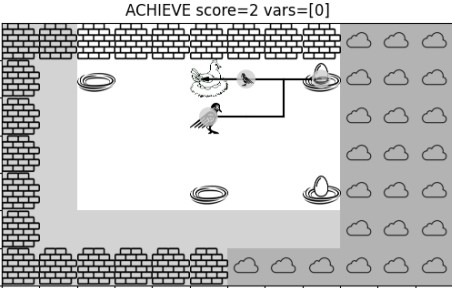

Figure 14: Food collection

- Level 5: Soccer level. In this level, as depicted in Figure 15, the agent needs to learn to shoot balls into the goal.

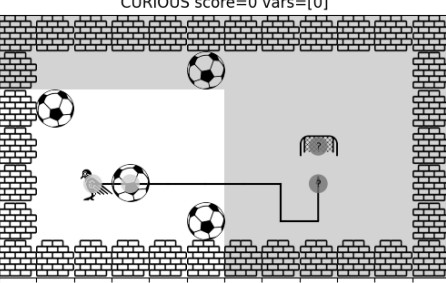

Figure 15: Food collection

- Level 6: Food collection while avoiding electric fences. In this level as depicted in Figure 16, the agent needs to collect food while avoiding electric fences.

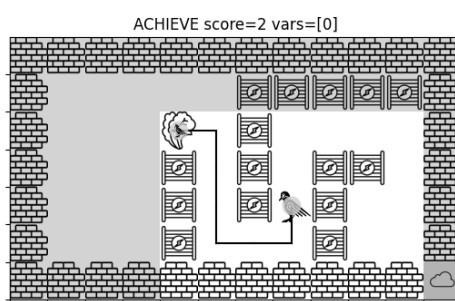

Figure 16: Food collection

- Level 7: Sokoban-like puzzle world (Dor & Zwick, 1999). In this level as depicted in Figure 17, the agent needs utilize the interaction properties of many different object types to successfully collect batteries:

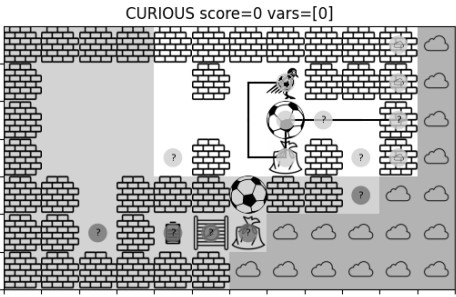

Figure 17: Sokoban-like puzzle world

