# OpenReview forum: "A grid world agent with favorable inductive biases"
_ICLR.cc/2025/Conference — Submitted to ICLR 2025_

### Official Review · Reviewer_yvs6 · 2024-10-29

**Soundness:** 3
**Presentation:** 4
**Contribution:** 3
**Rating:** 8
**Confidence:** 2

**Summary:**

The authors introduce NACE, a novel learning agent that utilizes a causality-informed curiosity model to make intelligent hypotheses about causal information in grid world environments. NACE is comprised of 4 components: an observer that updates a "bird-view" map of the environment and assesses prediction-observation failures, a hypothesizer that generates new rules, a planner that balances an exploration-exploitation tradeoff for accruing reward and refining hypotheses, and a predictor that models the environment. The authors assess NACE in a variety of environments from the Minigrid library clustered into three relevant groups: stationary environments, dynamic environments, and dynamic environments with sequential dependencies. Although NACE does not always find the optimal policy, its data efficiency is unparalleled by modern DRL algorithms.

**Strengths:**

- Existing RL techniques for solving gridworlds are systematically laid out and elaborated on in Section 3, which makes it easy for the reader to contextualize the work.
- Section 4 introducing NACE is concise well-described.
- Section 5 provides compelling results with a comparison to multiple baselines. Figures highlight the salient contributions that the authors attempt to make with NACE: extreme sample efficiency.
- The overall prose of the paper is extremely clear.

**Weaknesses:**

- A more thorough discussion of the 5 kinds of inductive biases, including examples, would make them easier to grasp.
- A diagram depicting the states and rule representations described in section 4.3 would be useful. Section 4.3 could use more development and examples.
- An example of a full set of causal rules for a simple environment would be welcomed.

**Questions:**

1. How were the hyperparameters chosen for the baseline algorithms?
2. Why is NACE unable to find the optimal policy? What improvements could be made to enable NACE to do so? A case-study on a specific environment would be interesting.

---

> ### Author Response · Authors · 2024-11-15
>
> We greatly appreciate the time and effort you invested in reviewing our work. We address your comments and suggestions, starting with answering your questions:
>
> Q1: Thank you for highlighting this. We plan to include the hyperparameter choices in the final submission. For most of the baseline algorithms, we adopted hyperparameters from their original repositories to ensure fair comparisons.
>
> Q2: We agree that this requires further discussion. Both theoretical considerations (e.g., NACE rules not capturing the statistics of the structure of generated environments) and practical factors (e.g., length of planning horizon which will benefit from more efficient implementation, and choice of exploration parameters such as the truth expectation threshold) contribute to NACE’s inability to consistently find the optimal policy, which we will elaborate on in the final version.
>
> Addressing weaknesses:
>
> W1: Agreed. We will expand this section to provide more detailed explanations for each inductive bias and how in particular it is realized in the technique.
>
> W2: We agree and will incorporate a diagram and additional examples for Section 4.3 to clarify the state and rule representations.
>
> W3: Thank you for this suggestion. We will include a causal rule in the final version to illustrate NACE’s structure more effectively.
>
> Thank you for your thoughtful review and valuable insights. We appreciate your engagement with our work and look forward to incorporating your suggestions to enhance the final version of our paper.

---

> > ### Author Response · Authors · 2024-11-27
> >
> > Thank you very much for your feedback. We have uploaded the newer version of the paper, incorporating your input. We plan to revise it even further prior to the deadline.
> >
> > - W1: How the inductive biases are utilized in NACE is now explained in detail in section 3.2.
> >
> > - W2: The state and rule representation section 3.1 now contains a corresponding diagram of the rule structure
> >
> > - W3: We included an example of the bird view map as well as list the rules learned in this environment, it can be seen in Appendix E.

---

### Official Review · Reviewer_QixP · 2024-11-01

**Soundness:** 3
**Presentation:** 2
**Contribution:** 3
**Rating:** 5
**Confidence:** 4

**Summary:**

The paper introduces Non-Axiomatic Causal Explorer (NACE), an agent optimized for grid world environments using causality-informed intrinsic rewards and inductive biases, including temporal and spatial modeling, to achieve data-efficient learning. Unlike most standard RL approaches, which require extensive training data, NACE efficiently learns policies in fewer steps by systematically exploring unfamiliar states. Experiments in MiniGrid scenarios show NACE's superior sample efficiency across various environments. The paper suggests that NACE’s principles could extend to more complex domains, promising advancements in data-efficient reinforcement learning.

**Strengths:**

The authors target a very important and interesting question: How to incorporate inductive bias into Reinforcement Learning and increase the data efficiency. Moreover, the method is compared to various other already established algorithms and tested with different examples.

**Weaknesses:**

Unfortunately, the reviewer cannot recommend the paper for publication at ICLR due to the following issues:

- The reviewer notes that while NACE’s systematic exploration of unfamiliar states is highlighted as its primary distinction from other RL methods, the incorporation of additional inductive biases defined in Section 4.1 remains unclear. Could the authors elaborate on how each bias is implemented within NACE’s framework? Additionally, conducting ablation studies on the contribution of each inductive bias would provide valuable insight into their individual impacts on performance.

- In the experimental results, the authors present rewards over time steps. Could the authors clarify how time steps are defined in this context? Specifically, are these time steps equivalent to RL framework iterations, with each time step representing the generation and evaluation of a potential solution?

- The reviewer suggests that comparing computational costs between algorithms would enhance the study's rigor. The current comparison lacks detail, as one time step in NACE may involve higher computational complexity than in other algorithms.

- In many of the RL frameworks tested, rewards remain stagnant for extended periods. If the results were examined at a finer scale, would smaller reward changes become visible, or does the mean reward remain consistently at zero?

- After the initial rapid increase in reward, NACE plateaus below the maximum attainable reward across all environments. The reviewer recommends exploring this behavior further and considering modifications to the algorithm that might enhance performance during the latter stages of learning. This could provide insights into whether additional mechanisms could support continued improvement toward optimal rewards.

**Questions:**

See Weaknesses

---

> ### Author Response · Authors · 2024-11-15
>
> Thank you for your thoughtful and detailed review! Your feedback has provided us with valuable insights and suggestions for improving our work, and we hope that our responses clarify the design and contributions of NACE. We address each of your points below:
>
> W1: In accordance with your feedback we will integrate more specific descriptions for each inductive bias in 4.1 as follows:
>
> - Temporal Locality: NACE builds rules based solely on the current and previous state.
>
> - Causal Representation: NACE's rules are structured as "(precondition, action) implies consequence".
>
> - Spatial Equivariance: learned rules can be applied at any location.
>
> - State Tracking: NACE explicitly keeps track of a bird's-eye view map by recording observations into it, updating the values that are within its observability window.
>
> - Attentional Bias: only rules that show a change from the previous to the current time step, or differ from the predicted value, are considered for rule formation, evidence updating, and prediction.
>
> Regarding ablation studies: we agree and will include a discussion on this in the main paper, referring to the appendix for more detail. For instance, if the agent cannot track observation locations, its performance often declines after picking up a key when the previously observed door is no longer in view. Also, if rules are tied to the specific locations where they were learned, they fail to generalize across locations, requiring the agent to relearn the same knowledge at each location, which greatly reduces sample efficiency. Furthermore, without attentional bias using the change and prediction mismatch sets, processing becomes significantly more resource-intensive due to the need to create and update a larger number of rules.
>
> W2: Yes as in RL framework iterations. The Minigrid environment uses the Gymnasium API, whereby in each timestep the agent gets one observation from the environment, performs one action, and receives the resulting reward given by the environment. We will clarify this in the text.
>
> W3: Thank you for pointing it out. NACE's computational complexity depends primarily on the asymptotics of its planning algorithm, which we mentioned to be a depth- and breadth-bounded Breadth-First Search. We are happy to explicitly include the asymptotics in the final version for clarity.
>
> W4: The behavior varies across techniques and environments considered, and the standard deviations are helpful to answer your question. For instance, in the first environment, Minigrid-Empty-16x16, RND's average reward is zero, but its non-zero standard deviation indicates some variation even at the end. DQN, on the other hand, shows both an average reward and standard deviation of zero, with values consistently near zero throughout the run.
>
> W5: Thank you for your excellent input! We ran a new experiment in accordance with your suggestion, with an increased truth expectation threshold of 0.85 for rule usage. We find it makes the system spend longer (due to 3 confirmations of a rule until exceeding the threshold) to take explorative actions before exploiting its rule base, which makes the system less greedy (more likely to find optimal ways to obtain reward) and could help improving performance in non-deterministic environments as well.
> And as noted in other answers, additional factors contribute to non-optimality: NACE's rule-based learning mechanism, does not capture the statistics of the structure of the generated environments that DRL policies can leverage. Additionally, constraints in NACE's prediction horizon and search breadth further contribute to suboptimal performance and would benefit from more efficient implementation. For instance if NACE's planning depth in MiniGrid-DoorKey-8x8-v0 is restricted to 8, we find that NACE does not succeed within the maximum time steps in about 50% of the episodes, leading to an average reward of only 0.48 on average, increasing the gap to the optimal solution (0.975) further.
>
> We are sincerely grateful for your detailed review and thoughtful feedback. Your insights have been instrumental in highlighting areas for improvement and have provided valuable guidance for strengthening our work. Thank you again for your time and engagement. We welcome any further questions or feedback you might have.

---

> > ### Comment · Reviewer_QixP · 2024-11-26
> >
> > Thank you for the detailed response, which addresses some of my concerns.
> >
> > W1: The reviewer appreciates that the authors agree with the feedback. However, the revised version does not include any of the proposed ablation studies or discussions. Given that the inductive bias presented appears to be tailored specifically to the Minigrid environment and may lack general applicability, the reviewer believes a detailed analysis is essential.
> >
> > W3: Beyond the asymptotic performance, the reviewer would like to see an evaluation of the computational costs of NACE compared to competing algorithms.
> >
> > W5: The response highlights that NACE's performance is highly sensitive to certain hyperparameter choices, with one sub-optimal choice leading to a significant drop in performance. This dependency warrants further exploration and should be explicitly addressed in the main paper.
> >
> > Additionally, it appears that none of the changes mentioned in the responses to this or other reviewers have been incorporated into the revised paper. As a result, the reviewer is unable to adjust their rating and maintains their original assessment that the paper is marginally below the acceptance threshold.

---

> > > ### Author Response · Authors · 2024-11-27
> > >
> > > Please see the updated version, we plan to revise it even further prior to the deadline.
> > >
> > > - W1: We include a discussion of the effects of the omission of the inductive biases in Appendix A.
> > >
> > > - W3: The asymptotics are now described in the main text in the planner description of section 3.4. Hardware setup and performance costs as exported from our runs will be added later today in the appendix.
> > >
> > > - W5: This is now analyzed in detail, we ran the model with different planning horizon hyperparameter choices to analyze how it affects performance in Appendix A.

---

> > > > ### Author Response · Authors · 2024-11-27
> > > >
> > > > Further update:
> > > > - W1: Ablation studies regarding inductive biases are now present in more detail in Appendix B.
> > > > - W3: Hardware and runtime costs are now included in Appendix F.
> > > >
> > > > Thank you again for your valuable feedback, which at this point we have mostly incorporated.

---

> > > > > ### Comment · Reviewer_QixP · 2024-11-28
> > > > >
> > > > > Thank you for providing the updated manuscript.
> > > > >
> > > > > While the revisions have significantly improved the paper, the reviewer finds that many results and sections still feel rushed.
> > > > >
> > > > > - Appendix F: This section lacks detail and clarity. Additionally, it seems questionable that PPO, A2C, and other algorithms all report a runtime of 1 hour for all runs. When introducing a new algorithm, comparing computational costs with existing frameworks is crucial. These comparisons should ideally be conducted on the same (or at least similar) hardware to ensure consistency. However, the authors have run some algorithms on CPUs, others on CPUs + GPUs, and even different CPUs and GPUs, making it difficult to draw meaningful conclusions.
> > > > >
> > > > > -  While the new Appendix B adds relevant information, it could benefit from a more quantitative approach. For instance, in B.2, calculating the binomial coefficient seems unnecessary, while in B.3, a figure or additional evidence to support the authors' claims would strengthen the section. On a positive note, the reviewer appreciates the study on the planning horizon depth included in another appendix.
> > > > >
> > > > > - Unfortunately, none of the appendix sections are properly referenced in the main paper. Given the considerable length of the appendix, proper referencing is essential to guide readers and maintain clarity.
> > > > >
> > > > >
> > > > > The reviewer also notes that is has not been further explored why NACE converges to a sub-optimal reward across all test cases, which remains a significant drawback. While the authors provided some explanations in their initial response, this issue has not been adequately resolved. From the reviewer’s perspective, this limitation should be explicitly discussed in the paper, along with potential solutions.
> > > > >
> > > > > Lastly, the new comparison with Dreamer V3 is a valuable addition, but it feels misplaced in a new section rather than being incorporated into the existing plots for consistency and readability.

---

> > > > > > ### Author Response · Authors · 2024-11-28
> > > > > >
> > > > > > We have updated the paper further and polished it significantly according to your feedback.
> > > > > > - The appendix sections are now properly referenced in the main paper.
> > > > > > - As you suggested we removed the special case of calculating the amount of possible rule preconditions with the binomials.
> > > > > > - The sub-optimality which primary cause we identified to lie in representational limitations is now mentioned in the main paper (end of section 4.3) and described in more detail in Appendix A, "Representational Limitations". Additionally as you noticed we clarified how hyperparameters can contribute further to sub-optimal results "Appendix A, Study of Reduced Planning Horizon".
> > > > > > - We acknowledge that our models have been ran on different hardware by different authors, with the implementations using different ML software frameworks with different degrees of GPU utilization efficiency. However as it is visible, NACE was running on the slowest hardware (CPU only), yet took the least amount of time. Nevertheless our paper focuses on the sample efficiency rather than computational efficiency of the implementations as we did not implement the techniques we compared with ourselves.

---

> > > > > > > ### Author Response · Authors · 2024-12-01
> > > > > > > **Just a reminder**
> > > > > > >
> > > > > > > Thank you again for your valuable feedback, we kindly ask to respond to our last message before the deadline.

---

> > > > > > > > ### Author Response · Authors · 2024-12-01
> > > > > > > >
> > > > > > > > For your convenience, a list of additional changes to address your feedback as found in the PDF:
> > > > > > > > - What a timestep exactly consists of has been described in Section 4.
> > > > > > > > - The inductive biases in NACE have been clarified in section 3.2.
> > > > > > > > - Appendix B.5 adds additional discussion of inductive biases in the DRL techniques.
> > > > > > > > - The comparison with DreamerV3 in Section 4.1, 4.2, 4.3 is coherently integrated in plots and tables, and not in a separate section anymore.
> > > > > > > > - The ablation studies in the appendix A have been enhanced with a reference from initiating discussion in Section 4.3.
> > > > > > > >
> > > > > > > > As you can see we have addressed most of your feedback, which proved very valuable to improve our paper,
> > > > > > > > and hope you find the last PDF to be more coherent and worthwhile for publication than the previous versions.

---

> > > > > > > > > ### Comment · Reviewer_QixP · 2024-12-01
> > > > > > > > >
> > > > > > > > > Thank you for providing the updated manuscript. The authors added a lot of additional information and the paper has now double the length of the original submission.
> > > > > > > > >
> > > > > > > > > Again the reviewer wants to mention that the revisions have significantly improved the original version of the paper, however some of the main points still remain:
> > > > > > > > >
> > > > > > > > > - NACE is not able to find optimal or near-optimal policies for all of the experiments. While this is now at least discussed within the paper, this behaviour does not indicate good generalizability to even more complex test cases as already for 2D minigrid the algorithm is not able to find the optimal policies for any of the environments considered. This significantly limits the contribution of the paper in the reviewers opinion.
> > > > > > > > >
> > > > > > > > > - Having said this, the reviewer acknowledges that NACE shows good results with regards to sample efficiency and outperforms here other algorithms.
> > > > > > > > >
> > > > > > > > > - Based on the reviewers comments the authors added a section on computational efficiency. While the reviewer is pleased that the authors addressed this comment despite their main focus on sample efficiency, this new section is not very convincing. After the comment that, 'it seems questionable that PPO, A2C, and other algorithms all report a runtime of 1 hour for all runs', the authors added approximately and on average but again no detailed run times were provided. It is not even clear if the authors average over different algorithms here (which they should not). Given that the NACE times are also given per seed and per environment, it is not clear for the reader if and how the different times should be compared.
> > > > > > > > >
> > > > > > > > > The reviewer has decided to keep their current score.

---

> > > > > > > > > > ### Author Response · Authors · 2024-12-01
> > > > > > > > > >
> > > > > > > > > > > Again the reviewer wants to mention that the revisions have significantly improved the original version of the paper, however some of the main points still remain:
> > > > > > > > > >
> > > > > > > > > > Thank you for your thoughtful feedback and for highlighting the improvements. Your insights and active engagement have been instrumental in refining our work!
> > > > > > > > > >
> > > > > > > > > > > It is not even clear if the authors average over different algorithms here (which they should not).
> > > > > > > > > > Given that the NACE times are also given per seed and per environment, it is not clear for the reader if and how the different times should be compared.
> > > > > > > > > >
> > > > > > > > > > We regret that this appendix section is not yet fully clear. If the paper is accepted, we promise to address this in the camera-ready version. Besides structural improvements of Appendix F, we will include the individual runtime data for each technique, utilizing the detailed runtime information already available in the OtherLogs folder provided in our contribution's zip file. This clarification will ensure a more precise runtime comparison and better alignment with your suggestions.

---

### Official Review · Reviewer_ea49 · 2024-11-03

**Soundness:** 3
**Presentation:** 3
**Contribution:** 3
**Rating:** 6
**Confidence:** 4

**Summary:**

The authors designed NACE (Non-Axiomatic Causal Explorer), a learning agent that incorporates a set of inductive biases that the authors consider to be important for an acting agent. These include causal relations, temporal locality, spatial equivariance, state tracking, and attentional biases.
The design of the agent is based on predicate rules that are proposed by the agent given the observations. The agent then plans to either explore rules (to collect new evidence about the rule) or maximize reward.
Finally, the authors test this agent in various scenarios of Minigrid and compare it against a wide range of (deep) RL agents. They show that in these particular scenarios, NACE is particularly sample efficient compared to the RL agents.

**Strengths:**

1. The paper is motivated by the importance of the inductive biases they propose to grid world environments. Thus, the authors proposed to study these by incorporating them all in their agent design. Finally, showing that these biases have a huge effect on sample efficiency.
2. The paper is mostly well written with some gaps in notation that I had a hard time following (see Questions)
3. The agent design seems to be novel in the way they instantiate the different biases based on predicate rules.

**Weaknesses:**

1. It is clear that NACE beats all the (deep) RL agents. However, given the comprehensive design, it is hard to understand where the benefit comes from. Perhaps ablating the effects of each inductive bias would be a good way to understand its contribution. Moreover, all RL agents considered are used in all experiments, but each one of them incorporates different biases that are incorporated in NACE. Perhaps grouping the RL agents based on the biases would make a clearer point of the importance of each bias.
2. RL baselines are shown to be less sample efficient. This could be the result of their generality (less inductive biases) as claimed. But I’m concerned that it seems that in all these cases the problems violate the Markov assumption, putting all these RL agents at a disadvantage. Is there an explicit handling of partial observability? Are there any RNNs/memory involved?
3. In the formal presentation of the agent, some notation is overloaded (e.g. c for cells, clauses in a rule, c(r) in line 288) which makes some of the method presentation hard to follow.
4. Although this is stated at the core of the paper, NACE is specifically designed for the grid world considered. It’s unclear how the results would extrapolate to other type of tasks. Also, I think it would be relevant to compare NACE to RMax, at least to discuss its similarities and differences.

**Questions:**

1. How does this compare to RMax? It seems to me that it has a similar flavor, in which we observe transitions and the agent explores such transitions until sure.
2. The formal definition of a cell is missing. I supposed the cell is the value of the 3rd dimension of the state definition.
3. Is there any value estimation happening? If so, how are you estimating the value function?

Minor comments
- Planner. Lines 311-314. Unclear wording.
- Overloading c(r) I think (line 288)
- Fix notations (use \citep)

---

> ### Author Response · Authors · 2024-11-14
>
> Addressed questions:
>
> Q1: The key differences lie in NACE’s compound state representation, which enables it to handle the high-dimensional state space of Minigrid efficiently, the particulars of how uncertainty is estimated, and the agent’s intrinsic motivation to seek out for states its knowledge applies the least in.
>
> Q2: In the state and rule representation section, we mention that the state is an array of cell values. As suggested, we will formalize and clarify this further. In the Minigrid environment, this array is provided for the observation window, and NACE integrates this into its "mental map", including both observed and currently unobserved cell values outside its view.
>
> Q3: NACE obtains the expected returns for the trajectories within its planning horizon as specified in the Planner description in section 4.4., which makes it choose one trajectory over the other. Hereby only the first action of the obtained plan is executed, then the cycle repeats.
>
> Addressed weaknesses
>
> W1: Thank you for this observation, we will add more information about each bias in relation to the techniques used.
>
> W2: The problems we addressed can be modeled as POMDPs. Partial observability was considered in most DRL techniques via LSTMs, as they were previously applied to partially observable Minigrid environments, enabling fair comparison. In NACE, partial observability is handled through its explicit map maintenance. We believe that addressing partial observability effectively remains an open research question, rather than an inherent limitation of NACE or our comparisons.
>
> W3: Thank you for pointing this out, we will correct this notational oversight to improve clarity.
>
> W4.1: We agree and have explicitly stated that NACE handles grid worlds, which remain challenging for RL methods and have many variations. Extending NACE to work outside of grid worlds (such as to continuous environments) would be an interesting direction for further research, as mentioned in our future works.
>
> W4.2: To our knowledge, RMax depends on an explicit tabular model with discrete, non-structural states, which does not scale to high-dimensional spaces like Minigrid's 2D-array-based observation window. Current literature indicates that DRL methods with intrinsic rewards are SOTA in Minigrid, which is why these methods have been the basis of our comparison, along with more basic DRL algorithms. However, we acknowledge that RMax’s optimistic value estimates could be valuable if extended to a DRL model. If you are aware of an RMax extension that could apply here, we would be glad to reference it in our final submission.
>
> Thank you for your valuable review. We hope that our responses clarify the design and contributions of NACE, as well as address any potential misunderstandings regarding its capabilities and limitations. We appreciate your thorough assessment and would be happy to engage further on any remaining questions.

---

> > ### Comment · Reviewer_ea49 · 2024-11-26
> >
> > Thank you for your answers and clarifications!
> >
> > I will be keeping my score the same. While I think the paper has some interesting contributions, I believe it would benefit from addressing these reviews in the manuscript.

---

> > > ### Author Response · Authors · 2024-11-27
> > >
> > > Please see the updated version, we plan to revise it even further prior to the deadline.
> > >
> > > - W1: We clarified the inductive biases in regards to NACE in section 3.2.
> > >
> > > - W2: The appendix now also contains information regarding hyperparameters and structure of the policies, including utilization of convolution layers and LSTM in Appendix D.
> > >
> > > - W3: The definition is now explicit and does not overload variables anymore. (Hypothesizer in section 3.4)
> > >
> > > - W4.1: We include discussion of how the system can be extended to operate beyond grid worlds in section 4.4.

---

> > > > ### Comment · Reviewer_ea49 · 2024-11-28
> > > >
> > > > Thank you for addressing my concerns. I've reviewed the updated manuscript and I appreciate the author's effort in incorporating the reviewers concerns. I'll be increasing my score.
> > > > Nonetheless, I was wondering if the authors could also clarify how the value estimation of the planner was done? Is it using Monte Carlo estimates (similar to MCTS)? TD? I believe it's relevant to readers to understand these details of the algorithm.

---

> > > > > ### Author Response · Authors · 2024-11-30
> > > > >
> > > > > Thank you for having re-evaluated the quality of our contribution; we are pleased to hear that our addressing of your concerns was satisfactory!
> > > > >
> > > > > In response to your inquiry: Unlike MCTS, which estimates state values by sampling rollouts that prioritize high-reward branches, NACE calculates the expected return over its planning horizon using Breadth-First Search. Exploring MCTS as an alternative presents a promising direction for addressing more complex environments, thank you for implicitly pointing us toward this. We will also ensure this is clarified further in the camera-ready version, should our paper be accepted.

---

### Official Review · Reviewer_euZ6 · 2024-11-04

**Soundness:** 3
**Presentation:** 3
**Contribution:** 2
**Rating:** 5
**Confidence:** 4

**Summary:**

The authors present NACE, a learning agent which uses strong inductive biases, causal reasoning and a causally-informed intrinsic reward to explore more efficiently in grid-world environments. NACE maintains an internal state consisting of a 2D array corresponding to each cell of the grid world, a 1D array to track non-spatial values such as inventory, as well as a set of rules of the form “(preconditions, action) => consequence” with counts of associated positive and negative evidence. At each step, it updates the 2D array and calculates which observed cells changed and which did not match their predicted values, uses this evidence to update the set of rules, then plans an action sequence to maximize expected return– or if no positive return trajectory is found, then to reach a state with minimum familiarity (average over all cells of how well they match the best fitting rule). Finally, the best-fitting rules are used to predict the cell values of the next state. They test on a number of minigrid environments and show that NACE reaches good performance in about 1000 steps, while existing DRL methods take around 1e6-1e7 steps to reach similar performance, although the best methods converge to higher average rewards at the end of training.

**Strengths:**

The sample efficiency results look very good.

 In general, the writing quality is high.

The Observer and Hypothesizer components of NACE, along with the State Match measure of state familiarity, appear to be quite novel.

Such a method should be quite interpretable - though the authors do not show any of the rules learnt by NACE in the test environments.

**Weaknesses:**

The authors do not mention or compare to existing methods for efficient structured learning which capture inductive biases, for example [1]. It is hard to evaluate the work’s originality given that the authors did not contextualize it among existing related approaches.

Though NACE heavily relies on an explicit model of the gridworld, they also do not compare to any explicitly model-based deep RL algorithms such as [2] or [3]

The significance of the contribution seems limited. NACE shares a lot of weaknesses with existing methods- (depends heavily on quality of state representations, would struggle where defining impactful state changes is difficult) - while lacking strengths (adaptable to continuous state spaces or high-dimensional action spaces, theoretical optimality guarantees). It seems limited to very simple rules, and the environments the authors tested on likewise covered a very small number of dynamics- navigating to a goal location with obstacles, and picking up a key to unlock a door to test sequential dependencies.

 - The authors did not test the ability to develop rules that capture dependencies across space rather than time, e.g. the need to flip a switch to unlock a set of doors. In fact, because the precondition constraints are defined on cells’ relative positions to the consequence cell, this method would likely do poorly on this dynamic, since this constraint would be best expressed as a condition on a cell specified by its global position (the switch location).

 - The constraints also require the cells to be exactly equal to a certain value, and are limited to cases where all constraints must be satisfied, rather than other conjunctions like Or, which excludes dynamics where values need only be above some threshold or within a set of allowable values (e.g. the Put Near minigrid environment where the agent must place one object near to another object).

 - The environments did not contain any stochasticity or objects that can move independently of the agent, e.g. the Dynamic Obstacles environment. A core component of NACE is observing which cells changed at each step and using that to create and update rules- is this method robust to settings where cells change irrespective of the agent’s action?


The clarity of the paper has room for improvement:
 - The cell notation is inconsistent and confusing- the subscript changes between $c$, $c_r$, $c_{t,x,y}$, $c_t$ without any explanation. Different symbols should be used for cell variables than for cell values e.g. in the definition $\bar{c}:=(c_r=c)$. If the precondition constraints are on cells’ relative positions, there should be notation for that in contrast to the global position notation $c_{t,x,y}$

 - K is used for the number of rules and also the number of equality constraints- consider using a different symbol.
some aspects of the method were not fully explained- see the Questions section.

 - Should consider using a different notation for the Match Quotient, since Q is usually used for the Q value function in RL.

 - Small grammar errors throughout the paper. E.g. “Such [an] approach” on line 154, quotation marks are flipped on line 163

[1] Tsividis, Pedro A., et al. "Human-level reinforcement learning through theory-based modeling, exploration, and planning." arXiv preprint arXiv:2107.12544 (2021).

[2] Hafner, Danijar, et al. "Mastering diverse domains through world models." arXiv preprint arXiv:2301.04104 (2023).

[3] Sekar, Ramanan, et al. "Planning to explore via self-supervised world models." International conference on machine learning. PMLR, 2020.

**Questions:**

Is the match quotient Q(r,c) defined for cell c being the consequence cell?

New rules are created “when positive evidence is found for the first time” - but how are the set of precondition equality constraints determined for the new rule? I.e., how does NACE determine which cells are relevant?

Why is positive evidence only counted for a rule if all of the precondition cells changed values and/or didn’t match the prediction at the last step? Since the precondition is an AND conjunction of many cell values, it is possible only one might need to change for a rule to be activated. And why can the positive evidence count still increase even if the rule fails to predict the outcome?

Why is the predicted reward not the sum, rather than the average, of the reward of each of the N utilized rules? Each rule seems to describe a way to obtain a certain reward, so if multiple rules are satisfied shouldn’t multiple rewards be obtained?

---

> ### Author Response · Authors · 2024-11-14
>
> Thank you for your thoughtful questions and observations!
>
> Addressing questions:
>
> Q1: Yes it is. We agree it may not be entirely clear from the current wording. We will make this explicit in the final version and appreciate your feedback.
>
> Q2: The relevant cells are determined by the union of the two sets: change set, and prediction mismatch set. (1) The change set contains cells that have been changed from the last time step to the current step. (2) The prediction mismatch set contains cells that have a different observed value than was predicted for the current time step. While this is already part of the formalization of $w_+(r)$, we can mention this explicitly to enhance clarity.
>
> Q3.1: It is part of the conceptual design to increase the computational efficiency by considering only cells that have either changed or ("or" due to the union of the sets in the definition of w+(r)) differ from the prediction as "relevant cells" for rule formation, evidence updating, and selective prediction.
>
> Q3.2: The match quotient measures to what degree the rule preconditions match the observed cell values. It equals to 1 only if all the precondition cell value constraints of the rule are satisfied.
>
> Q3.3: It cannot, positive evidence is only obtained when the rule predicts correctly. The misconception might stem from the fact that the rule formation is only considering cells in the set of changed cells and prediction mismatch cells, which acts as a filter that is separate from the actual cell values to compare.
>
> Q4: There are usually multiple rules utilized to predict a given state in its entirety. In the simplest case, if the reward prediction of all these rules aligns with the observed reward, their average will also align, while the sum would overestimate the outcome.
>
> Addressing weaknesses:
>
> W1: While we believe our approach introduces novel formulations not present in existing literature, we understand the importance of contextualizing it with related work in structured learning. Our Related Works section currently includes comparisons with several relevant methods, but we will add the reference to further highlight distinctions and similarities.
>
> W2: NACE does not rely on an pre-defined model of the grid world. It starts with empty rule base, building them purely from observations, without assumptions about the environment beyond its strong priors. Only after these rules are learned, the agent knows how to operate in the environment. The rules are derived directly from the observation arrays provided by the environment, with no predefined notion of the objects involved.
>
> W3: Rules in NACE are based on evidence measures rather than true/false values, providing some tolerance to less precise state representations. We acknowledge that comprehensive studies are needed to demonstrate and quantify this. Regarding lacking strengths, Grid World environments remain a challenge in RL, for multiple reasons, such as the sequential dependencies, as you mentioned. We do not consider the use of rules itself as a limitation, provided they effectively capture relevant dependencies and enable the agent to learn efficiently and perform competently.
>
> W3.1: True, but it falls outside of the scope of the tested domains. Extending the work to learn rules based on absolute coordinate values is reserved for future work, and not essential for Minigrid benchmarks.
>
> W3.2: Since the agent can learn multiple rules with AND conditions, it can capture many cases where OR conditions might otherwise be required. For example, ((a OR b) implies x) is logically equivalent with ((a implies x) and (b implies x)). As for dynamics involving variable values, this is outside the scope of the Minigrid benchmarks.
>
> W3.3: NACE was specifically designed to handle "agent-external changes" within the observation window of the agent, as hypotheses formed from spurious correlations produce wrong predictions, generating negative evidence.
>
> W4.1: We appreciate your suggestion and will incorporate this change for improved clarity.
>
> W4.2: Thanks for catching this. We will correct the symbol usage and clarify related areas of confusion in the final version.
>
> W4.3: With "Match Ratio" R conflicts with the reward symbol. "Match Goodness" with the symbol G may be an alternative.
>
> W4.4: Thank you for noting. We will certainly correct these issues.
>
> We are grateful for your thorough and insightful review, which helps us strengthen our work. And we hope our responses have clarified potential misunderstandings!
> We would also like to highlight that this is a new technique and the first paper to comprehensively compare it with DRL in grid worlds. We hope these comparisons add value for the research community as well.

---

> > ### Comment · Reviewer_euZ6 · 2024-11-21
> > **Response to authors**
> >
> > Thank you for your response.
> >
> > >positive evidence is only obtained when the rule predicts correctly.
> >
> > I don't see that in the definition of when $w_+(r)$ is updated to $w_+(r)+1$ from line 266 to line 277- there is no mention of checking that the value of $c_t$ matches any prediction.
> >
> > >NACE does not rely on an pre-defined model of the grid world
> >
> > I said explicit, not fully pre-defined. I am referring to the 2-dimensional array where each value explicitly represents one cell in the grid world (which I guess does have to be pre-defined with knowledge of the size and shape of the grid world, though the details of this appear to be missing in your paper). **My concerns about the lack of comparison to model-based deep RL methods still stand.**
> >
> > >W3.3: NACE was specifically designed to handle "agent-external changes" within the observation window of the agent, as hypotheses formed from spurious correlations produce wrong predictions, generating negative evidence.
> >
> > What if the agent-external change is non-random and it would be helpful to predict it? It seems that NACE would be very inefficient at learning to predict such changes, because all the rules include an action. Thus, multiple copies of the agent-external change rules would be created for every action the agent might take.
> >
> >
> > Regarding the responses arguing that X weakness is outside the scope of Minigrid: this says more about the simplicity of Minigrid as a benchmark than the value of the approach. How does this work provide useful generalizable knowledge to the research community if it is so anchored to this specific simplistic benchmark, and will require extensive manual modification to work for less toy environments?

---

> > > ### Author Response · Authors · 2024-11-24
> > >
> > > "I don't see that in the definition of when is updated to from line 266 to line 277- there is no mention of checking that the value of matches any prediction."
> > >
> > > We apologize for the notational oversight and appreciate you pointing it out. Indeed, the equality constraints of the postconditions must hold for positive evidence to increase.
> > >
> > >
> > > "I said explicit, not fully pre-defined. I am referring to the 2-dimensional array where each value explicitly represents one cell in the grid world (which I guess does have to be pre-defined with knowledge of the size and shape of the grid world, though the details of this appear to be missing in your paper). My concerns about the lack of comparison to model-based deep RL methods still stand."
> > >
> > > You are correct that one grid cell corresponds to one cell in the agent's map representation. However, the size and shape of the agent's map do not need to be pre-defined, as the memory array can dynamically grow. This allows the agent to operate without prior knowledge of the world's dimensions.
> > >
> > >
> > > "What if the agent-external change is non-random and it would be helpful to predict it? It seems that NACE would be very inefficient at learning to predict such changes, because all the rules include an action. Thus, multiple copies of the agent-external change rules would be created for every action the agent might take."
> > >
> > > You are correct that a separate rule is required for each action to predict agent-external changes. However, in environments with a limited number (typically fewer than 20) of discrete actions, NACE remains sample-efficient. To explore agent-external changes, World 5 in the provided source package models a Pong-like game in a grid world, where NACE successfully learns to predict the ball's movement. In future work, NACE could be extended to learn action-independent rules, with such rules accumulating negative evidence due to prediction failures if the outcome depends on the chosen action.
> > >
> > >
> > > "How does this work provide useful generalizable knowledge to the research community if it is so anchored to this specific simplistic benchmark, and will require extensive manual modification to work for less toy environments?"
> > >
> > > In addition to our comprehensive comparison of model-free DRL techniques in grid worlds, our contribution extends beyond the NACE architecture itself to introduce principles that apply to more complex environments:
> > >
> > > - The curiosity model, encourages an agent to seek unfamiliar states where its knowledge is least applicable (via precondition match, differing from traditional predictability measures in the literature), while also maximizing the likelihood of reaching these states.
> > >
> > > - The concept of lazily updating representations, focusing primarily on observed changes and prediction mismatches, rather than exhaustively updating all representations.
> > >
> > > - The principle of compositionality in states and predictions, where each rule predicts only part of a state, and multiple rules combine to predict the full state.
> > >
> > > - Count-based evidence accumulation, which efficiently tracks prediction success and failure, considering both the prediction success ratio and amount of contradictions and confirmations of rules.
> > >
> > >
> > > "My concerns about the lack of comparison to model-based deep RL methods still stand."
> > >
> > > We acknowledge that comparisons with model-based deep RL techniques could be valuable in the future. However, we think that such comparisons are not currently aligned with the State-of-the-Art in this domain. Recent advancements have demonstrated significant improvements in sample efficiency through the inclusion of intrinsic rewards—an approach closely related to our own formulation of Curiosity. This guided our selection of comparison techniques, focusing on methods that reflect these advancements.
> > >
> > > If you have a specific model-based technique in mind, we would greatly appreciate a concrete suggestion to address your feedback constructively.
> > >
> > >
> > > “Regarding the responses arguing that X weakness is outside the scope of Minigrid: this says more about the simplicity of Minigrid as a benchmark than the value of the approach.”
> > >
> > > While we did not propose Minigrid ourselves, we note that it is a widely respected benchmark in the DRL community, valued for addressing critical challenges such as partial observability and sequential dependencies. These characteristics have motivated our investigations into relevant inductive biases, which are not limited to Minigrid environments, even though are helpful in the domain.
> > >
> > > Thank you again for your detailed and valuable feedback!

---

> > > > ### Comment · Reviewer_euZ6 · 2024-11-25
> > > >
> > > > Thank you for your further clarifications.
> > > >
> > > > >The concept of lazily updating representations
> > > >
> > > > What are these representations? Do you mean updating rules?
> > > >
> > > > >our contribution extends beyond the NACE architecture itself to introduce principles that apply to more complex environments
> > > >
> > > > You have provided no evidence that these principles will still be applicable/helpful in more complex environments. E.g. in high dimensional spaces, it seems likely that preconditions won't match for almost all unvisited states, making this an unhelpful form of curiosity. The principle of compositionally is also not novel, and it is not surprising that it applies to Minigrid which was designed that way.
> > > >
> > > > >Recent advancements have demonstrated significant improvements in sample efficiency through the inclusion of intrinsic rewards
> > > >
> > > > Intrinsic rewards may help sample efficiency, but they can be added to model-based methods too- they are not inherently tied to model-free RL. Sample efficiency is well-known to be one of the main advantages of MBRL over Model-free RL. Considering that your method is model-based and uses curiosity, the most fitting comparison would be a state of the art model-based deep RL method with the addition of curiosity-based intrinsic rewards. As I mentioned earlier, Dreamer [1]  is a well known state of the art MBRL algorithm.
> > > >
> > > > [1] Hafner, Danijar, et al. "Mastering diverse domains through world models." arXiv preprint arXiv:2301.04104 (2023).

---

> > > > > ### Author Response · Authors · 2024-11-25
> > > > >
> > > > > Thank you for your active engagement in the discussion, which we highly appreciate! Please find our responses below:
> > > > >
> > > > > > What are these representations? Do you mean updating rules?
> > > > >
> > > > > Indeed, the rules we create and the mechanisms for selectively updating their evidence.
> > > > >
> > > > > > You have provided no evidence that these principles will still be applicable/helpful in more complex environments. E.g. in high dimensional spaces, it seems likely that preconditions won't match for almost all unvisited states, making this an unhelpful form of curiosity.
> > > > >
> > > > > We agree that a direct application of our curiosity model in high-dimensional spaces would face challenges. However, this limitation stems not necessarily only from our model but also from the need for complementary mechanisms to reduce high-dimensional continuous states to manageable, lower-dimensional discrete representations. Techniques such as ConvNets for object detection or other feature abstraction methods are already addressing similar issues of space reductions for numerous domains and we expect it can address our concerns and enable our principles to generalize to more complex real-world environments.
> > > > >
> > > > > Importantly, selective processing in NACE enhances computational efficiency in larger grid worlds, assuming most cell values remain static between time steps. However, computational efficiency was not our primary focus.
> > > > >
> > > > > > The principle of compositionally is also not novel, and it is not surprising that it applies to Minigrid which was designed that way.
> > > > >
> > > > > While Minigrid states are compositional by design, our method extends on this by predicting entire states using competing rules for each cell and applying only the winner. This winner-takes-all approach, based on the Cell match value definition, extends beyond grid-based compositions and can generalize to more complex state representations.
> > > > >
> > > > > > Intrinsic rewards may help sample efficiency, but they can be added to model-based methods too- they are not inherently tied to model-free RL. Sample efficiency is well-known to be one of the main advantages of MBRL over Model-free RL. Considering that your method is model-based and uses curiosity, the most fitting comparison would be a state of the art model-based deep RL method with the addition of curiosity-based intrinsic rewards. As I mentioned earlier, Dreamer [1] is a well known state of the art MBRL algorithm.
> > > > >
> > > > > Thank you for pointing us to this very interesting reference! We hence drafted a comparison with DreamerV2 which fits into the space from reducing our lenghty descriptions of the compared techniques in section 3 as another reviewer suggested.
> > > > >
> > > > > We now compare the advantages and drawbacks of the explicit state transition (full state) and rule representation (partial state) with those of learning a latent dynamics model. The latter approach, as employed by DreamerV2, offers broader applicability beyond grid worlds and is well-suited for domains with high-dimensional state spaces.
> > > > >
> > > > > At the same time, we show quantitative comparisons for some of the environments, whereby we show the converged performance (first table) as well as when NACE reaches the performance DreamerV2 converges to (second table):
> > > > >
> > > > > | Technique | Environment | Average Reward | Time Steps |
> > > > > |-----|-----|-----|-----|
> > > > > | NACE | DoorKey-6x6-v0 | 0.93 | 1.00E+05 |
> > > > > | DreamerV2 | DoorKey-6x6-v0 | 0.89 | 1.00E+05 |
> > > > > | NACE | Unlock-v0 | 0.86 | 2.50E+06 |
> > > > > | DreamerV2 | Unlock-v0 | 0.60 | 2.50E+06 |
> > > > >
> > > > > | Technique | Environment | Average Reward | Time Steps |
> > > > > |-----|-----|-----|-----|
> > > > > | NACE | DoorKey-6x6-v0 | 0.89 | 3.00E+02 |
> > > > > | DreamerV2 | DoorKey-6x6-v0 | 0.89 | 1.00E+05 |
> > > > > | NACE | Unlock-v0 | 0.60 | 2.70E+02 |
> > > > > | DreamerV2 | Unlock-v0 | 0.60 | 2.50E+06 |

---

> > > > > > ### Author Response · Authors · 2024-11-27
> > > > > >
> > > > > > Thank you for your valuable feedback, which has greatly influenced the revisions to our paper. We have addressed several shortcomings of the initial submission, leading to significant improvements:
> > > > > >
> > > > > > 1. Updated Results: The noted tables with new results are now included in Section 4.4, along with an extended discussion. This section is fully dedicated to comparing our approach to Dreamer in the context of Model-based RL.
> > > > > >
> > > > > > 2. Improved Motivation: The introduction has been revised to better motivate our approach and clarify its positioning within related works.
> > > > > >
> > > > > > 3. Broader Generalization: We have expanded on how our ideas could generalize beyond grid worlds in section 4.4.
> > > > > >
> > > > > > We plan further revisions before the deadline and welcome additional feedback.

---

> ### Comment · Reviewer_euZ6 · 2024-11-27
>
> Thanks for your responses.
>
> >Techniques such as ConvNets for object detection or other feature abstraction methods are already addressing similar issues of space reductions for numerous domains and we expect it can address our concerns and enable our principles to generalize to more complex real-world environments.
>
> The additional training time for the feature abstraction methods such as CNNs must then be taken into account, reducing or potentially nullifying the method's sample efficiency advantage. Furthermore, the features extracted by a general-purpose feature abstraction model might not be the most fitting ones to plug into your method, which could harm both sample efficiency and final performance. These claims of applicability to more complex environments would be more convincing if the authors could provide evidence with an empirical demonstration.
>
> >We hence drafted a comparison with DreamerV2
>
> I appreciate the fast turnaround for this, but why did you compare with DreamerV2? I cited DreamerV3 - DreamerV2 is no longer state of the art. I also find it odd and concerning that you ran it on different environments than the ones used for the other deep RL model comparisons- why DoorKey-6x6-v0 instead of DoorKey-8x8-v0? The standard deviations are also missing. Instead of presenting the comparison in two tables, it would be much more informative while also saving space to add the full learning curves to your existing Figures 2-7 along with all the other baselines. I don't think it is necessary or helpful to separate this comparison into a separate section.

---

> > ### Author Response · Authors · 2024-11-27
> >
> > **Regarding training time for the feature abstraction methods:**
> >
> > Thank you for your insightful comments! We agree that the sample efficiency of methods relying on feature abstraction can be diminished when the abstraction model is trained concurrently with the agent’s policy. Methods like Dreamer, which leverage gradient-based updates to improve representations dynamically, indeed hold an advantage in this respect. We now already highlight this point more explicitly in our comparisons to provide a balanced perspective.
> >
> > Furthermore, in many real-world applications, the relevant object types and features are often predefined, with models pre-trained accordingly. Hybrid systems that leverage such pre-training are common in industry. however we acknowledge that they typically demand large engineering effort.
> >
> > **Regarding comparison with DreamerV3:**
> >
> > Thank you for your invaluable feedback! Initially, we compared our approach with DreamerV2 since it is peer-reviewed and has multiple published results in Minigrid, whereas DreamerV3 is currently a pre-print. Also, we encountered challenges with running DreamerV3 on our HPC cluster in Minigrid, as it lacks a standard Gymnasium interface, unlike the techniques we compared against and DreamerV2.
> >
> > However we totally agree and understand the importance of using the state-of-the-art version and in the meanwhile have identified comparable numbers for DreamerV3 in the literature. We have hence included an initial comparison with DreamerV3 in the current version.
> >
> > Additionally, we have now managed to run DreamerV3 on the environments ourselves on our HPC cluster, however the runs are not yet fully completed. Once the results are available, we will integrate them into the plots and tables as you suggest, hopefully by the end of this day. Lastly, we will ensure standard deviations are included for completeness and consistency, as is already the case for the preliminary comparison in the current version.
> >
> > Thank you for your proactive and constructive feedback, it has helped making our paper stronger! We want to let you know that most of your feedback is now integrated, as you can see in the recent revision, and hope it will be sufficient for you to reconsider the rating.

---

> > > ### Author Response · Authors · 2024-11-28
> > >
> > > We updated the PDF further. As promised DreamerV3 results are now seamlessly integrated with the comparison section exactly as you desired. Not all runs are plotted to convergence yet, as our HPC runs are still ongoing (SimpleCrossing, Unlock, DoorKey), but will be complete in our last revision prior to the deadline. Thank you very much for having encouraged us to strenghten our comparison!

---

> > > > ### Author Response · Authors · 2024-11-29
> > > >
> > > > For your convenience, the final version addresses additional points of your feedback:
> > > > - Integration of DreamerV3 into Section 4 with proper result interpretation as you suggested.
> > > > - Expanded references to model-based techniques in Related Works (Section 2.1).
> > > > - Discussion of agent-external changes and non-determinism in Appendix A (Representational Limitations).
> > > > - Improved discussion about DreamerV3 and NACE’s strengths and weaknesses in Section 4.3.
> > > > - Further clarifications of notations with a table of all notations in Appendix C.
> > > > - Match quotient renamed to 'M' to avoid confusion with Q-values in Section 3.3.
> > > >
> > > > Thank you for considering our updated version, and we hope this list is helpful!

---

> ### Comment · Reviewer_euZ6 · 2024-11-30
>
> Thanks for your response. The integration of the DreamerV3 results in section 4 looks good. The discussion of limitations and better contextualization with more relevant related works is also valuable. I think the soundness of the paper has been significantly improved, and thus I raise my score.
>
> Unfortunately, I still cannot recommend acceptance since I still do not see sufficient concrete evidence that the proposed methods can actually be useful beyond Minigrid. Currently, I am doubtful that the contribution is sufficient for ICLR.

---

> > ### Author Response · Authors · 2024-12-01
> >
> > Thank you for having re-assessed our paper and for further engaging in valuable discussions!
> >
> > We appreciate your suggestion to explore extensions beyond grid worlds, aligning closely with our ongoing work.
> > Specifically, we have been developing a ROS 2 node to adapt our method to continuous environments faced by mobile robots, which involves:
> > - Use of a SLAM algorithm to maintain an occupancy grid map.
> > - Downsampling the grid map to match the robot's dimensions upon each map update.
> > - Applying YOLO for object detection on the camera feed and mapping bounding boxes with depth information to grid cells.
> > - Providing NACE with this semantic grid to operate similarly to its behavior in simulation.
> > - Sending NACE's movement actions to Nav2 for navigation and MoveIt2 for pick and drop operations.
> > This integration enables NACE to handle navigation, exploration, and pick-and-place operations in real-world scenarios.
> >
> > We wonder if this extension meets your expectations.
> > In case it does, we could incorporate these results into the appendix for the camera-ready submission including a link to a working demo?

---

> > > ### Comment · Reviewer_euZ6 · 2024-12-02
> > >
> > > The described extension could provide such evidence, if it showed that NACE works well compared to existing methods, without requiring extensive modification or tailoring to the new environment. However, the description of the extension by itself does not constitute evidence, and at this stage in the process it may be inappropriate to propose such a significant extension, and for a reviewer to change their score based on such an extension.

---

### Official Review · Reviewer_Saa1 · 2024-11-04

**Soundness:** 3
**Presentation:** 2
**Contribution:** 3
**Rating:** 6
**Confidence:** 3

**Summary:**

NACE (Non-Axiomatic Causal Explorer) is a novel experiential learning agent leveraging causal reasoning and intrinsic reward signals to enable more efficient learning within grid world environments. The authors compare the proposed method against state-of-the-art RL algorithms, demonstrating its benefit in terms of sample efficiency across many different grid world environments.

**Strengths:**

- Novelty: the work brings novelty due to the adoption of a curiosity model based on causal reasoning.
- Narration: the paper's narration is well-done and sound, and the work is generally well-written.
- Experiments: the experimental campaign is convincing since it considers several state-of-the-art RL algorithms and exploration frameworks. The evaluation metric regards the sample efficiency of each method, demonstrating NACE's brilliant results.
- Supplementary materials: the attached zip file containing NACE's codebase runs easily and smoothly.

**Weaknesses:**

- **Some notations are not very clear.** In particular, the section dedicated to the NACE architecture (section 4.4) leaves some symbols unexplained, such as the observer's sets $M_t^{change}, M_{t}^{observation-mismatched}, M_t^{prediction-mismatched} $, which have been introduced here only in mathematical notation. Still, I would suggest to explain their meaning. Same for the function $f_{exp}$ whose usage and terms composition are not completely clear.
- Apart from the notation, also **intuitions behind the need for some components of the architecture are not immediately understandable**. I would have rather added an appendix to explain those details more deeply. For example, I would explain the interactions between the different components of the architecture more verbosely, also describing the flow diagram in Figure 1 and the role of each component in natural language, to give an intuition about the maths behind it. Perhaps, a pseudocode of the entire algorithm could come in handy.
- The main limitation of NACE is due to its application since it is **usable only in deterministic grid world settings**. However, authors highlight as future works possible extensions to more complex problems.
- **Experimental setups could have been explained more in detail** in the Appendix, by reporting a more extended description of the presented scenario, perhaps with the support of the relative images (bird-view map). Furthermore, authors could add those scenarios that have not been presented in the main paper, but that can be run in the codebase, such as the *soccer world*.
- **Hardware employed to run the experiments and time consumption of the framework** not provided.

**Questions:**

- From learning curves is evident that NACE is more sample efficient than all the other tested algorithms. However, I would like to ask why it is not able to reach the optimal policy and which can be the intuition behind this recurrent behavior.
- Thinking out of the grid world environment, I would like to ask how this method can work and if you see limitations and challenges that have to be considered in more complex problems.
- Regarding non-deterministic transitions, how can NACE give "system tolerance" as stated in line 294?

---

> ### Author Response · Authors · 2024-11-14
>
> Thank you for valuable feedback and your interesting questions!
>
> We address each question in the following paragraphs:
>
> Q1: Besides possible implementation limitations, NACE rules do not exploit the statistics of the generated environment structure, which DRL policies do capture. Additionally, limits in the prediction horizon and search breadth bounds in the planning component of NACE can lead to more greedy, non-optimal behavior, even when relevant knowledge is already learned, as the expected return is calculated over the planning horizon.
> We appreciate that you noticed the data efficiency properties, which is NACE's core strength. Our paper also outlines sample efficiency differences across the compared DRL techniques, which we believe could provide additional value to the research community.
>
> Q2: Complexity can increase in various ways. For instance, what NACE delivers is the effective consideration of sequential dependencies, which, as we also demonstrated, significantly increases the training data demand of Deep Reinforcement Learning techniques, even with techniques that involve intrinsic rewards for better sample efficiency. This can also be seen in our paper when going from MiniGrid-Unlock-8x8-v0 to MiniGrid-DoorKey-8x8-v0 (Lines 486-505), which only adds one necessary additional sequential dependency to reach a goal location after the door has been opened with the key. Our DRL results in this regard are also consistent with the results in the literature.
> A key challenge, as you noted, is generalizing beyond grid worlds. We are currently researching NACE's integration with mobile robots in simulation using ROS2, featuring automatic downsampling of occupancy grid maps, and continuous action invocation via Nav2. However, this work lies outside the scope of the current paper.
>
> Q3: NACE achieves system tolerance through continuous evidence updating. The truth expectation of a rule determines whether it is being utilized, and this is a statistical measure that takes the prediction success rate into account as well as how many data points met the rule preconditions. With increasing amounts of collected evidence, the truth expectation measure becomes more stable, making the system to consider the statistically most likely outcomes of its actions even when they do not always generate the same outcome.
>
> Now, we address the weaknesses:
>
> W1: Respectfully, we agree in part and recognize that the page limit restricted us from providing extended explanations for some mathematical notations. However, we feel that these notations are not entirely ``unexplained", since the description of the Observer includes a brief sentence about these sets. We will clarify these descriptions for improved readability.
>
> W2: Thank you for this suggestion. We agree that additional explanations would enhance understanding. We will include a more detailed appendix covering interactions among architecture components as well as the pseudo-code of the algorithm, which we already have, but could not fit in the main body of the paper.
>
> W3: With respect, we believe the statement ``usable only in deterministic grid world settings" is somewhat inaccurate. While we focused on deterministic domains to maintain consistency with the nature of the RL benchmarks used for comparison, our technique is explicitly designed to handle non-determinism via evidence accumulation and updating. Specifically, the $f(r)$ value of a rule reflects its prediction success ratio, which stabilizes (and its truth expectation increases) with accumulated evidence $w(r)$, as noted in our prior response to question 3.
>
> W4: Thank you for your suggestion, we will include experimental setups and additional information in the appendix for the final version.
>
> W5: Thank you for pointing it out, we have hardware information available and we will include it too.
>
> Thank you for your detailed review and valuable feedback. We hope to have addressed your questions and suggestions thoroughly, and we are working to incorporate these improvements in the final version of the paper. We appreciate your insights, which will help strengthen our work, and we hope our ideas sparked your interest. Please let us know if there are any further questions or if additional clarifications are needed.

---

> > ### Comment · Reviewer_Saa1 · 2024-11-26
> >
> > Dear authors,
> >
> > thank you for addressing my comments.
> >
> > As other reviewers have highlighted, I don't see a revised version of the paper addressing our concerns. Even if your work is valuable, for future submissions I suggest you pay more attention to presentation and notation consistency, which have been weak points of your paper.
> >
> > For these reasons, I won't change my rating - marginally below the acceptance threshold.

---

> > > ### Author Response · Authors · 2024-11-27
> > >
> > > Please see the updated version, we plan to revise it even further prior to the deadline.
> > >
> > > - W1: We significantly improved the textual explanation of the sets in the Observer description in section 3.4.
> > >
> > > - W2: The flow diagram now comes with a textual description in section 3.4 Observer description. Additionally, pseudocode of NACE can now be found in Appendix C.
> > >
> > > - W3: In Appendix A we already include a basic analysis of the system's ability to handle non-determinism including quantitative analysis in an example environment.
> > >
> > > - W4: We included an example of the bird view map as well as list the rules learned in this environment in Appendix E. We might also add illustrations of the test environments in the final submission as you suggest.
> > >
> > > - W5: Exact hardware information will be included in our last revision prior to the deadline, including time consumptions as we measured for running the models. We assume you are fine with it being in the appendix.

---

> > > > ### Comment · Reviewer_Saa1 · 2024-11-27
> > > >
> > > > Thanks for all the work done on the updated version of the paper. Since clarifications have been made and readability improved, I'll update my score to "marginally above the acceptance threshold."

---

> > > > > ### Author Response · Authors · 2024-11-27
> > > > >
> > > > > Thank you for your reconsideration and the valuable feedback you provided that allowed us to improve our contribution!

---

### Official Review · Reviewer_WhYe · 2024-11-14

**Soundness:** 2
**Presentation:** 2
**Contribution:** 1
**Rating:** 6
**Confidence:** 2

**Summary:**

The authors propose NACE, a technique to efficiently solve grid world environments and compare this to state-of-the-art deep reinforcement learning algorithms.

**Strengths:**

The experimental results are easy to follow, and the figures are well made.

**Weaknesses:**

I find the motivation of the authors' interest in gridworld problems to be lacking, and the testbeds to be simplistic. I am not convinced that RL is unable to solve such simple tasks as is claimed by the authors, and believe this to be due to suboptimal hyperparameters which appear to be missing from the text. The overall presentation of the paper lacks sufficient depth in details to where it is difficult to follow along in a meaningful way with notation left undefined. It is written in a manner not meant to be read by someone seeing this material for the first time. For example, Subsection 4.3 should likely be in the beginning of Section 4 or at least before Section 4.2, as it formally defines a rule, what a cell is, that you are doing conjunction, etc. All of this we can assume in 4.3, but for clarity, it should be clearly stated beforehand.

Some areas the authors spend too much time explaining - for example, DQN or PPO, and a whole page is dedicated to these algorithms; each algorithm's description/shortcoming should have been reduced to 1-2 sentences (e.g., don't need to define DQN here just get to the point), giving the authors 0.5 page back that could have been used to better explain their contribution. At the end, I am left with a feeling that this is nothing new, I am still unclear how this compares to existing work *that is similar*, and how everything ties together. Also, how can DQN not solve MiniGrid-Empty-16x16-v0 but can solve MiniGrid-DistShift2-v0? This makes me question hyperparameters, because it should have been possible for DQN to randomly discover at least once a path from start to goal and then improve upon it, like is seen in the other more difficult task.

Many notations are not defined in 4.4 to the point where the paper is frustrating to read. What are c, v, a, R(r). Is lower-case r rules? Or reward? Why a lower-case r for a set of rules?

**Questions:**

1. Where does the Curiosity Model fit into the overall NACE Architecture in 4.4?
2. What are c, v, a, R(r). Is lower-case r rules? Or reward? Why a lower-case r for a set of rules?

---

> ### Author Response · Authors · 2024-11-15
>
> We appreciate your review and the feedback provided! Below, we respond to your observations and outline adjustments.
>
> Questions:
>
> Q1: The Curiosity Model is a key component of the planner in NACE. It operates as a secondary objective in planning, guiding the agent to systematically explore states where its existing knowledge is least applicable. This is quantified through the State Match Value, derived from the match quotients of rules applied to cells in the state. During planning, if no action sequence leads to greater-than-zero expected return, the planner uses the Curiosity Model to identify action sequences that minimize the State Match Value, encouraging the agent to explore less familiar areas. It supports NACE’s systematic acquisition of missing causal knowledge.
>
> Q2: Thank you for raising this. In our notation, $c$ represents a cell, $v$ is a value, $a$ denotes an action, and $R(r)$ refers to the reward predicted by a rule $r$. We chose concise lower-case letters to align with the variables’ names, but we recognize that this may cause confusion given the multiple variables introduced. We will revise and clarify the notation.
>
> Weaknesses:
>
> Motivation: We acknowledge that grid worlds may not fully reflect the complexity of real-world environments, and therefore may not align with everyone’s research interests. However, they are valuable for testing and benchmarking algorithms, particularly in addressing challenges such as partial observability, sequential dependencies, and combinatorial state spaces. Minigrid, in particular, is a widely used benchmark and has been featured in numerous recent publications at ICLR and other leading conferences. Additionally, many SOTA techniques were extensively studied in Minigrids, including those used in our comparisons, further underscoring its relevance and the open challenges it presents. While we recognize that grid worlds are not inherently realistic, their controlled settings enable systematic experimentation and comparison, which aligns with our research goals. We are happy to elaborate on this motivation to provide greater context.
>
> RL in grid worlds: With respect, we do not intend to claim that RL is incapable of solving these tasks. If our wording suggests otherwise, we are happy to revise it. Some of our results demonstrate competent performance from various DRL techniques, which we trained and comprehensively evaluated across the environments, and these results are largely consistent with findings in the literature. However, we acknowledge that some of the weaker results may stem from hyperparameter choices, which we adopted from the repositories cited in the corresponding papers. We appreciate your suggestion to include them, as this will enhance the quality of the final submission.
>
> Missing details: We agree that Subsection 4.3 before 4.2 would improve clarity, as state is not yet defined in its current order. Thank you for noting this oversight. This adjustment will resolve the issue and improve clarity, particularly for first-time readers engaging with these novel concepts.
>
> Lenghty DRL descriptions: We greatly appreciate this observation, as it provides the space we need to address the other suggestions and further improve the paper.
>
> Lack of novelty: We are optimistic that the novelty of our contribution can be objectively assessed, as our architecture is unique and built on formulations not found in existing literature. However, we acknowledge the importance of strengthening the Related Works section to better contextualize our approach and highlight its distinctive aspects.
>
> Performance variations: Note that we are reporting results within a maximum of 10^7 time steps. DQN is not learning on MiniGrid-Empty-16x16-v0 in this time frame because the problem space is significantly larger than in MiniGrid-DistShift2-v0. Given additional time steps, DQN would certainly be able to learn in MiniGrid-Empty-16x16-v0 as well. For context, we also tested MiniGrid-Empty-6x6-v0, which has a similar problem size to MiniGrid-DistShift2-v0, and DQN converged within (9 * 10^5) time steps. This is expected, as the task is simpler and demonstrates how DQN's performance is influenced by the problem size and time steps allotted.
>
> Undefined notations: As we mentioned in our response to a related question, $c$ represents a cell, $v$ denotes a value, $a$ refers to an action, and $R(r)$ is the reward associated with a rule $r$. Lower-case letters were chosen for alignment with their respective names, but we understand your frustration and will hence spend time to improve our notations.
>
> We appreciate the time and effort you dedicated to reviewing our paper. Your feedback has highlighted areas where clarity and structure can be improved, and we are committed to addressing these in the final version. Specifically, we will refine our notations, reorganize sections for better flow, and adjust our explanations to ensure the content is accessible and engaging for readers.

---

> > ### Comment · Reviewer_WhYe · 2024-11-26
> >
> > Thank you for your response and for answering my questions/concerns.
> >
> > A quick thought - one issue with using symbol $r$ to refer to a rule (although it makes sense) is often in RL the immediate reward is also $r$. Perhaps a suggestion might be to simply write out $rule$ to make the distinction more clear?
> >
> > Unfortunately I will still leave my rating unchanged - marginally below the acceptance threshold. Simply because I am unable to see the revised version addressing my (and the other reviewers') concerns with the presentation; a more succinct explanation with clear notation will greatly improve the reading, as well as placing the novelty front and center - emphasizing even more how it addresses the existing gaps in literature.

---

> > > ### Author Response · Authors · 2024-11-27
> > >
> > > Please see the updated version, we plan to revise it even further prior to the deadline.
> > >
> > > - Missing details:
> > > We fixed the section ordering and enhanced the description of states in the states and rule representation section which is now 3.1.
> > >
> > > - Lenghty DRL descriptions: We have significantly reduced this section which is now section 2.1.
> > >
> > > - Novelty: We strenghtened introduction, related works (particularly 2.1) and compare now also with an additional technique in section 4.4.
> > >
> > > - Performance variations:
> > > Hyperparameter choices are now documented in Appendix D, including performance influence of NACE planning horizon.
> > >
> > > - Undefined notations:
> > > Besides resolving ambiguities in chosen notations (section 3.3, 3.4), we have added a table in Appendix B which describes all notations. Switching from $r$ to $rule$ was a promising suggestion however it made the formalizations too lengthy. Consequently the notation choice is now outlined in the notation description and we are now consistently only using capital $R$ for reward.

---

> > > > ### Comment · Reviewer_WhYe · 2024-11-27
> > > >
> > > > Thanks for all your work in revising the submission. I looked over the article's changes and it is definitely improved from initial draft. I have updated my score to "marginally above the acceptance threshold". Is there a particular area or statistic referencing the number of rules typically required to solve each of these environments? I may have overlooked this, but I cannot find one. This seems important if NACE is rule-based.

---

> > > > > ### Author Response · Authors · 2024-11-27
> > > > >
> > > > > Thank you for the score update and for strengthening our contribution!
> > > > >
> > > > > > Is there a particular area or statistic referencing the number of rules typically required to solve each of these environments?
> > > > >
> > > > > We do not pre-define the particular rules to identify, as they are dynamically created. And they are spatially independent which greatly reduces the amount of constructed rules. In this regard, we observe that most Minigrid environments demand a rule number of about 50, usually less than 100 as now also indicated in the Appendix H.

---

> > > > > > ### Comment · Reviewer_WhYe · 2024-11-27
> > > > > >
> > > > > > Thank you for pointing me in the right direction for this; 50-100 seems reasonable. This may be a less pressing matter, but if accepted, it would be fantastic to provide a clearer picture on the rule count with mean, standard deviation, confidence intervals etc for each experiment task. However, for the time-being, please continue your efforts addressing any concerns the other reviewers might have.

---

> > > > > > > ### Author Response · Authors · 2024-11-27
> > > > > > >
> > > > > > > Thanks for the prompt response! We will spend the remaining time to incorporate other reviewer's feedback trying to get some extra points to get our paper accepted. In case the paper gets accepted we will ensure to incorporate rule counts and associated statistics within the camera-ready version of the paper.

---

### Author Response · Authors · 2024-12-03
**Summary**

We thank the reviewers for their constructive feedback and active engagement in discussion, which significantly strengthened our contribution.

Our submission, "A Grid World Agent with Favorable Inductive Biases," presents three major contributions:

1. A comprehensive comparison of 11 DRL techniques across 6 Minigrid environments, evaluating sample efficiency and performance.
2. A detailed discussion on inductive biases that significantly enhance sample efficiency in grid world environments.
3. A novel experiential learning agent, NACE, designed for grid-world environments with incorporated inductive biases.

**Reviewer Feedback:**
Reviewers praised the novelty of the formalisms, value of the inductive biases and the comprehensive experimental setup for comparison.
Initial suggestions for improvement included:

- Enhanced clarity in notation and conceptual explanations.
- Clearer documentation of individual inductive biases and their implementation.
- Addressing NACE's convergence and generalizability beyond grid-world environments.
- Including comparisons with a model-based RL technique.
- Reporting computational costs and hardware specifications for the experiments.

**Author Responses and Revisions:**
We provided detailed responses to each review and made several key revisions:
- Expanded explanations of the inductive biases in the main text and appendices.
- Improved notation, added diagrams, and reorganized sections for better readability.
- Detailed discussions on NACE's performance limitations and potential improvements.
- Updated the manuscript with computational cost details and hardware specifications.
- Included comparisons with DreamerV3, a state-of-the-art model-based RL method.

Impact and Contribution:
While some reviewers had reservations about the general applicability regarding point 3 of our contribution (NACE),
the overall sentiment indicates that the work contributes valuable insights into sample-efficient learning through the included inductive biases.
Following revisions, four reviewers increased their scores, reflecting the improvements made.

We believe these enhancements substantiate the manuscript’s contribution.
Should the paper be accepted, we plan to further refine it,
including detailed runtime metrics for each technique in the camera-ready version.

---

### Meta-Review · Area_Chair_BCJd · 2024-12-19

**Metareview:**

This paper presents NACE (Non-Axiomatic Causal Explorer), a novel experiential learning agent leveraging causally-informed intrinsic reward for grid world environments. While reviewers acknowledged the paper's contribution to sample-efficient learning and thorough empirical validation across multiple environments, significant concerns were raised about the method's fundamental limitations. Specifically, NACE fails to achieve optimal performance even in simple grid worlds, shows high sensitivity to hyperparameter choices, and lacks convincing evidence for generalizability beyond grid-world domains. Through extensive revisions, the authors improved notation clarity and added comparisons with state-of-the-art methods like DreamerV3, but core concerns about theoretical foundations and broader applicability remain insufficiently addressed.

Given these substantial limitations and the preliminary nature of the current results, I recommend rejection with encouragement to strengthen the theoretical framework and demonstrate effectiveness in more complex domains.

**Additional Comments On Reviewer Discussion:**

The paper exhibits three critical unaddressed concerns:

1. The method's generalizability remains unproven beyond the specific domain of grid worlds, with no empirical evidence supporting its applicability to more complex environments.
2. Despite its sample efficiency, NACE consistently converges to sub-optimal policies across all test environments, raising questions about its fundamental limitations and practical utility.
3. The empirical validation lacks standardization in computational comparisons and thorough analysis of the method's scalability, particularly in relation to state-of-the-art approaches.

---

### Decision · Program_Chairs · 2025-01-22

Reject